# Productivity loss associated with functional disability in a contemporary small-scale subsistence population

Jonathan Stieglitz[1,2]*, Paul L Hooper[3], Benjamin C Trumble[4,5], Hillard Kaplan[3], Michael D Gurven[6]

[1]Université Toulouse 1 Capitole, Toulouse, France; [2]Institute for Advanced Study in Toulouse, Toulouse, France; [3]Economic Science Institute, Chapman University, 1 University Drive, Orange, United States; [4]Center for Evolution and Medicine, Life Sciences C, Arizona State University, Tempe, United States; [5]School of Human Evolution and Social Change, Arizona State University, Tempe, United States; [6]Department of Anthropology, University of California, Santa Barbara, Santa Barbara, United States

**Abstract** In comparative cross-species perspective, humans experience unique physical impairments with potentially large consequences. Quantifying the burden of impairment in subsistence populations is critical for understanding selection pressures underlying strategies that minimize risk of production deficits. We examine among forager-horticulturalists whether compromised bone strength (indicated by fracture and lower bone mineral density, BMD) is associated with subsistence task cessation. We also estimate the magnitude of productivity losses associated with compromised bone strength. Fracture is associated with cessation of hunting, tree chopping, and walking long distances, but not tool manufacture. Age-specific productivity losses from hunting cessation associated with fracture and lower BMD are substantial: ~397 lost kcals/day, with expected future losses of up to 1.9 million kcals (22% of expected production). Productivity loss is thus substantial for high strength and endurance tasks. Determining the extent to which impairment obstructs productivity in contemporary subsistence populations improves our ability to infer past consequences of impairment.

**\*For correspondence:**
jonathan.stieglitz@iast.fr

**Competing interests:** The authors declare that no competing interests exist.

## Introduction

As the longest living and slowest growing primate, humans experience unique physical impairments with potentially large economic and social consequences. Human hunter-gatherers have lower adult mortality than chimpanzees and a significant post-reproductive lifespan (*Gurven and Kaplan, 2007*; *Hill et al., 2007*; *Kaplan et al., 2010*; *Kaplan et al., 2000*; *Wood et al., 2017*), which generates ample opportunity for incident degenerative disease and dysfunction (*Finch, 2010*). Evidence of greater dietary reliance by hominins on difficult-to-acquire resources including hunted game at least two mya (*Aiello and Wheeler, 1995*; *Antón et al., 2014*; *Thompson et al., 2019*; *Ungar, 2012*) suggests delayed peak skill acquisition in adulthood (*Gurven et al., 2006*; *Kaplan et al., 2000*; *Koster et al., 2020*; *Walker et al., 2002*), possibly when senescence was already underway. Functional disability of degenerative origin may hinder hominin foraging in costly ways. These costs amplify if disability hinders resource transfers or provisioning of other assistance, given their potential fitness impacts (*Gurven et al., 2012*; *Hawkes, 2003*; *Hill and Hurtado, 2009*; *Hooper et al., 2015*; *Marlowe, 2003*; *Schniter et al., 2018*; *Wood and Marlowe, 2013*). The ubiquity among hunter-gatherers to form social groups of clusters of multi-generational resource-pooling units (*Kaplan et al., 2009*; *Migliano et al., 2017*), and in base camps at least ~400 kya (*Kuhn and Stiner,*

*2019*), may reflect a species-typical strategy of complex cooperation to minimize daily risks, including production shortfalls. Quantifying the cost of disability in terms of productivity loss in extant small-scale subsistence populations is thus critical for understanding the strength of selection pressures for cooperative strategies that minimize risks associated with disability.

Skeletal evidence suggests that functional disability of degenerative origin may not have been that uncommon throughout hominin evolution, and that participation in activities essential for survival and reproduction (e.g. bipedal locomotion, food acquisition, and load carrying) may have been constrained by disability. For example, marked thoracic kyphosis and vertebral disc degeneration are evident in *Australopithecus afarensis* (AL-288) from ~3.2 mya (*Cook et al., 1983*), as is spondylolisthesis in middle Pleistocene humans (SH-1) from ~430 kya (*Bonmatí et al., 2010*; also see *Trinkaus, 2018*). These ailments regularly cause lower back pain, difficulty walking, and carrying or lifting objects among modern clinical patients (*Francis et al., 2008*), who may provide indirect insights into the consequences of disability in past hominin populations. Given the variability across hominin species in terms of environmental exposures, diet, life history traits (e.g. lifespan), morphology (e.g. body and brain size) and locomotion from ~4 mya until 40 kya (*Wood and Boyle, 2016*), whether and how these impairments hindered survival and reproduction is unclear, and not readily discernable from any single hominin species.

Joint behavioral and epidemiological study of contemporary small-scale subsistence populations suggests that diverse impairments of degenerative or other etiology hinder resource production and transfers. Among Yora forager-horticulturalists of Peru, men are unable to forage due to illness or injury on 11% of all days (*Sugiyama and Chacon, 2000*). Among Shiwiar forager-horticulturalists of Ecuador, >60% of individuals experience an impairment (e.g. chronic pain, fracture, laceration, infection, animal bite or sting, and burn) severe enough to interfere with subsistence work for ≥1 month; bone fractures – a focus of the current study – may be more likely than other impairments to cause prolonged disability (*Sugiyama, 2004*). Among Tsimane forager-horticulturalists of Bolivia, the population studied here, 75% of adults report being bedridden due to illness or injury at least once in the 3 months prior to survey, and Tsimane are incapacitated on about 10% of all days (*Gurven et al., 2012*). The fact that these studies include both younger and older adults (aged 20+ years) indicates that prolonged disability is not conditional on reaching advanced ages.

While providing evidence that disability may have important consequences, the observational studies of contemporary small-scale subsistence populations mentioned earlier are limited: they rely mainly on self-reports and thus cannot identify objective, specific somatic impairments associated with diminished subsistence effort; they utilize crude measures of subsistence involvement (e.g. able to work or not) and thus cannot quantify the magnitude of productivity loss or ascertain the range of subsistence behaviors associated with disability; they often rely on small sample sizes of adults, including adults in their peak productive years; they do not consider potential for bidirectional associations between somatic condition and economic production; and they are not designed to isolate specific mechanisms linking somatic condition and productivity. Precise assessment of the economic cost of disability and consideration of potential compensatory strategies to mitigate disability-related productivity loss is important in light of the comparatively low mortality rates characteristic of human life histories (*Gurven and Kaplan, 2007*). Studies of contemporary small-scale subsistence populations can thus complement bio-archaeological and paleoanthropological studies for understanding factors affecting foraging behavior and resilience in past hominin populations. In addition, in populations lacking formal institutional mechanisms that minimize risks of economic insecurity (e.g. workers' compensation and disability insurance) studies of the economic cost of disability are important for understanding relationships between security, sociality, and well-being. As subsistence populations like the Tsimane increasingly participate in local market economies over time, studies of the economic cost of disability may also help inform policies designed to provide social security benefits.

Hominin social organization in multi-generational, multi-layered cooperative units are believed to be a necessary precursor for lowering mortality rates (*Bird et al., 2019*; *Hawkes et al., 1998*; *Hill et al., 2007*; *Isaac, 1978*; *Kaplan et al., 2000*; *Washburn and Lancaster, 1968*). The fact that many subsistence skills require cumulative knowledge that is often transmitted from older to younger generations implies that functionally disabled adults can maintain 'fitness value', despite perhaps compromised ability to produce and transfer resources (*Gurven et al., 2020b*). How functional disability may have influenced behavior is unclear. Some anthropologists have interpreted human

skeletal remains showing potential signs of impairment as evidence of compassion and morality of group members, who presumably would have had to support the disabled (see *Dettwyler, 1991* and references therein; *Tilley, 2015*). This interpretation is contentious but supported by observations of modern hunter-gatherers frequently sharing food with impaired adults (e.g. *Gurven et al., 2000*). Ethnographic reports of death-hastening behaviors among frail elders (e.g. abandonment), who themselves may advocate such behaviors (e.g. *Balikci, 1970*), suggest that decisions to support the disabled may be complex and influenced in part by their expected future productivity. Quantifying the economic cost of disability is therefore critical for understanding the extent of cooperation that must have co-evolved with hominin life history traits.

While there are diverse factors affecting functional ability, here we examine among Tsimane adults whether compromised bone strength is associated with diminished participation in routine subsistence tasks. We utilize thoracic computed tomography (CT) to measure two primary indicators of bone strength in thoracic vertebral bodies: bone mineral density (BMD), which accounts for ~70% of the variance in bone strength (*NIH Consensus Development Panel on Osteoporosis Prevention, Diagnosis, and Therapy, 2001*), and fracture presence and severity. We focus on thoracic vertebrae for several reasons. Thoracic vertebral deformity is regularly observed in diverse hominin skeletal samples (*Chapman, 1972*; *Cook et al., 1983*; *Dequeker et al., 1997*; *Foldes et al., 1995*; *Gresky et al., 2016*; *Lieverse et al., 2007*; *Mays, 1996*; *Trinkaus, 1985*). Thoracic vertebral fractures are also among the most common fragility fractures in humans living in post-industrialized populations (*Sambrook and Cooper, 2006*), and are more common among Tsimane compared to age- and sex-matched US (Los Angeles) controls (*Stieglitz et al., 2019*). While other great apes experience degenerative disease (*Jurmain, 2000*; *Lovell, 1990*), spontaneous vertebral fractures have not been observed in other apes, even in individuals with severe osteopenia (*Gunji et al., 2003*). Bipedal hominins thus appear to be especially susceptible to fragility fractures of the spine (*Cotter et al., 2011*) and perhaps other anatomical regions. Human thoracic vertebrae track mechanics of both lower and upper limbs, and thoracic vertebral body fracture can directly impede mobility. Clinical vertebral deformities are commonly associated with pain and impaired quality of life and can have serious long-lasting economic consequences (*Francis et al., 2008*). Declining thoracic vertebral body BMD is a manifestation of senescence more generally, and rather than directly inhibiting specific functional abilities per se, a co-occurrence of reduced BMD and disability could indicate more general degenerative processes that do not necessarily involve vertebral fracture (e.g. vertebral disc degeneration, osteophytes, and nerve damage). Regardless of the specific mechanisms, determining whether and how compromised thoracic vertebral body strength is associated with diminished productivity in an extant small-scale subsistence population whose behavior can be directly studied improves the ability to infer more generally potential economic costs of disability for an obligate biped reliant on extractive foraging and food sharing.

In this paper we identify economic correlates of compromised bone strength (i.e. thoracic vertebral body fracture and lower thoracic vertebral body BMD) by examining involvement in four common tasks in foraging economies: hunting, tree chopping, tool manufacture (i.e. weaving), and walking long distances. These four tasks were selected because of their importance in daily subsistence and their variation in strength and skill requirements. Hunting has likely been an important source of hominin protein and fat production since early *Homo* (*Aiello and Wheeler, 1995*; *Antón et al., 2014*; *Wrangham and Carmody, 2010*), and perhaps earlier, although the dietary availability of meat and other animal products (e.g. marrow) in the Plio-Pleistocene was variable across space and time. Tree chopping is essential for constructing shelters and footbridges for water crossings, hunting arboreal or burrowing prey that hide in tree trunks, manufacturing certain tools, gathering firewood, and horticultural production. Tool manufacture and repair have likely long improved extractive foraging efficiency for omnivorous, manually dexterous hominins, as suggested by the fact that Oldowan tool-using hominins absorbed energetic costs of transporting stones for processing carcasses over long distances (>10 km) (*Braun et al., 2008*). Weaving of items used for carrying diverse objects (e.g. woven bags that store animal carcasses) and for resting (e.g. ground mats) is routinely performed by women in numerous small-scale subsistence populations, including the Tsimane (*Figure 1*). Lastly, the ability to efficiently travel long distances is evident for bipedal hominins relative to quadrupedal apes by at least the mid-Pliocene (*Pontzer et al., 2009*); long-distance travel is crucial for participating in diverse hominin foraging tasks (e.g. hunting and fishing) and in social activities (e.g. visiting) within and across residential settlements.

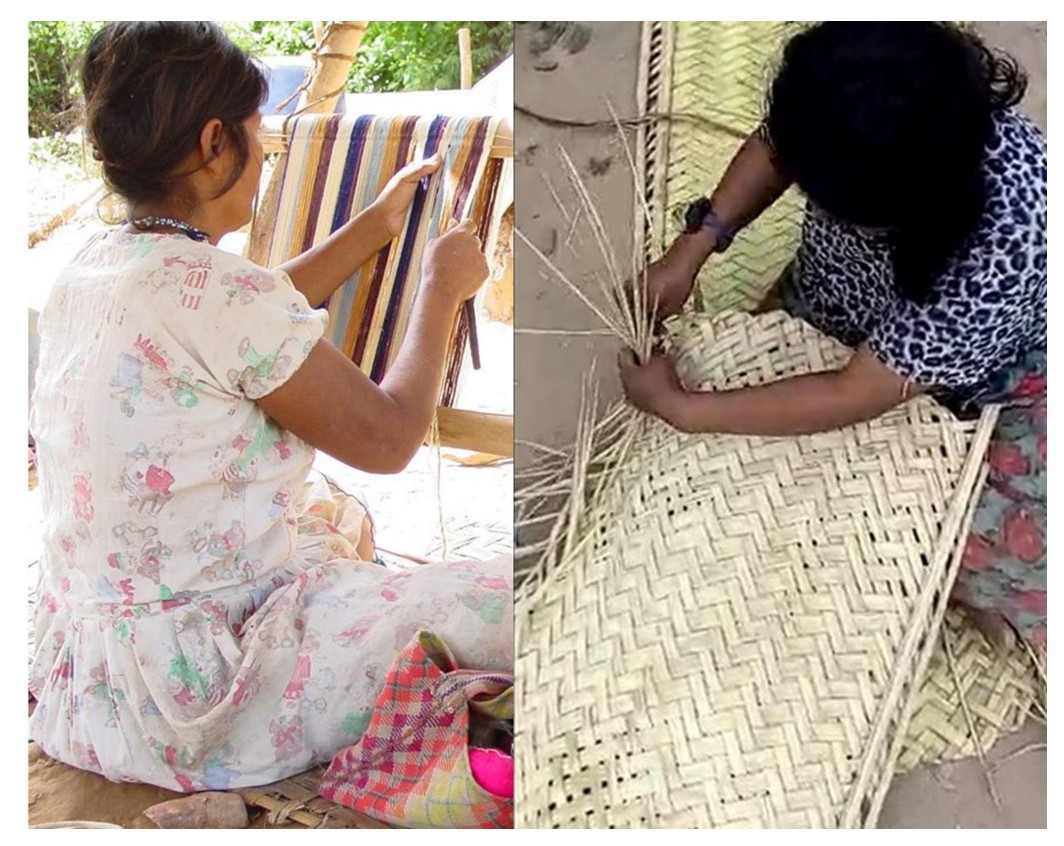

**Figure 1.** Tsimane women weaving bags used for carrying diverse objects (left panel; photo credit: Jonathan Stieglitz) and ground mats used for resting (right panel; photo credit: Arnulfo Cary). Finished woven products are also shown in each panel.

We first test whether probability of completely ceasing to perform a given subsistence task increases with compromised bone strength. There are several possible explanations for how compromised bone strength may be associated with reduced economic productivity, and here we focus on the mediating role of functional disability. One might expect considerable productivity loss associated with compromised human thoracic vertebral body strength, as most activities of daily living (e.g. sitting, walking, running, lifting, and even breathing) generate vertebral loads (*Myers and Wilson, 1997*; *Polga et al., 2004*; *Rohlmannt et al., 2001*; *Stewart and Hall, 2006*). For many common activities (e.g. neutral standing, standing with weight, mild trunk flexion and extension, and lifting objects above the head) the greatest spinal compressive loads are generated in the thoracolumbar region (*Bruno et al., 2017*). Compromised thoracic vertebral body strength could especially curtail participation in high-strength and/or high-endurance tasks involving greater vertebral loads, such as hunting and tree chopping; participation in less physically demanding tasks involving reduced vertebral loads, such as weaving, may not be strongly curtailed. To the extent that men may more often participate in tasks requiring high levels of strength and/or endurance in subsistence economies like that of the Tsimane with marked sexual division of labor, compromised thoracic vertebral body strength could disproportionately hinder male subsistence activities.

We next estimate the magnitude of productivity losses (i.e. lost kcals/day) from hunting cessation associated with compromised bone strength. Productivity losses are estimated in two ways: (i) daily losses at age $x$; and (2) expected cumulative future losses from age $x$ onward, where future losses are discounted by mortality. Substantial cumulative future productivity losses may be expected from compromised vertebral body strength because trabecular bone losses during adulthood are not fully regained (*Mosekilde, 1989*; *Mosekilde et al., 1987*). Once spinal fracture occurs it may impede daily subsistence activities for the remainder of life. We focus on hunting productivity loss given the centrality of animal-based foods to theoretical models of hominin behavioral and life history

evolution (*Bliege Bird et al., 2001*; *Hawkes et al., 2001*; *Hill and Hurtado, 2009*; *Isaac, 1978*; *Kaplan et al., 2000*; *Marlowe, 2000*; *Washburn and Lancaster, 1968*). Since hominins as a taxonomic group show high plasticity in life history traits (e.g. expected adult lifespan) and diet (e.g. reliance on meat), it is possible that our estimates of productivity loss associated with compromised bone strength better approximate those for hominins exhibiting relatively modern life histories. In the present context, hunting productivity losses refer to lost hunting calories from hunting cessation associated with thoracic vertebral body fracture and/or lower thoracic vertebral BMD.

We do not assume ex ante that compromised bone strength per se is the only cause of disability or reduced productivity, or that these factors necessarily co-vary in some coordinated fashion – these are empirical questions that are partially addressed in this paper. We do expect that individuals with compromised bone strength who are no longer able to perform specific subsistence tasks will adjust their time allocation depending on various factors (e.g. own somatic condition and household demand). To gain insights into potential compensatory behavioral strategies of adults with compromised bone strength who can no longer hunt, we compare expected productivity losses from hunting to observed fishing production rates, as a means of ascertaining the degree of foraging modification required to offset disability-associated energetic deficits. This exercise provides insights into whether adults with compromised bone strength who cease hunting, through their own behavioral modifications, can easily retain their ability to provision themselves and kin with critical animal products in the diet, or whether provisioning by group members or other behavioral modifications may be necessary. Lastly, utilizing data on anthropometrics and self-reported reasons for ceasing to perform specific subsistence tasks, we examine whether compromised bone strength precedes task cessation, rather than the reverse.

## Materials and methods

### Study population

Tsimane are semi-sedentary forager-horticulturalists inhabiting >90 villages in lowland Bolivia. Villages are usually located along rivers or other water sources (e.g. lagoons) and are composed of dispersed clusters of kin-related households. The household is the basic economic unit for production and consumption, and most daily coordination is confined to the nuclear and extended family. Tsimane hunt with a diverse toolkit that can include any/all of the following: rifle or shotgun (but guns and ammunition are often unavailable), bow and arrow, machete, slingshot and trap, and sometimes assistance from tracking dogs (*Gurven et al., 2006*). On average, men hunt about once every week or two; hunting is more common during the wet season and in interior forest villages. The average hunt lasts 8.4 hr and covers 17.9 km (*Trumble et al., 2014*) and often involves carrying animal carcasses long distances. Tsimane women occasionally hunt, much less often than men, and use different strategies (e.g. women rarely use firearms) with lower return rates . Because of women's relatively infrequent participation in hunting, we focus here only on men's hunting behavior. Fishing typically occurs in rivers, streams, or lagoons and involves use of hook-and-line, bow and arrow, machete, plant poison, and/or net. Single-day fishing and hunting activities are mostly solitary or in pairs with siblings, sons, in-laws, or age-mates. Entire families sometimes go on multi-day fishing and hunting trips that can last from 2 days to several months. Food sharing is widespread within extended families, but also limited in scope across families. Kinship and relative need largely determine the magnitude and direction of food transfers (*Gurven et al., 2012*; *Hooper et al., 2015*).

Much of the protein and fat consumed by Tsimane come from hunting and fishing (*Kraft et al., 2018*), whereas most (62%) of the total calories in the diet come from cultigens grown in small swiddens. In order to clear space for cultigens, machetes are first used to clear smaller vegetation, and then metal axes (and more recently chainsaws when available) are used to cut larger trees before planting and burning (*Trumble et al., 2013*). While both sexes plant cultigens, help clear smaller vegetation, and harvest, the felling of larger trees is typically done by men. Women's physical activity level (PAL; defined as the ratio of total daily energy expenditure to daily basal metabolism) is in the 'moderate to active' range (PAL = 1.7–1.9) and is constant throughout adulthood. Men's PAL is 'vigorously active' (PAL = 2.0–2.2) and declines by 10–20% from the peak (in the late 20s) to older adulthood (age 60+ years) (*Gurven et al., 2013*).

Physical strength peaks in the mid-to-late 20s, then declines thereafter, reaching ~60% of maximum adult capacity by age 70 (*Gurven et al., 2006*). Impaired musculoskeletal function is not uncommon among Tsimane adults (see Figure 3H in *Gurven et al., 2020a*). BMD estimates from the axial and appendicular skeleton – particularly from sites rich in trabecular bone (i.e. thoracic vertebrae, calcaneus, and distal radius) – suggest that osteoporosis is not uncommon, especially for postmenopausal women (*Stieglitz et al., 2015a*; *Stieglitz et al., 2016*; *Stieglitz et al., 2017*; *Stieglitz et al., 2019*). Early and rapid reproduction may contribute to women's skeletal fragility later in life. Compared to more sedentary Los Angeles matched controls, age-standardized prevalence of thoracic vertebral body fracture is two times higher for Tsimane women and 3.3 times higher for men using directly comparable CT methods (*Stieglitz et al., 2019*). For adults aged 50+, 38% experience some movement restriction (e.g. during flexion or extension), most commonly in the dorsolumbar, hip, knee, or shoulder regions. Nearly 70% of adults aged 50+ are also clinically diagnosed with some condition of the musculoskeletal system and connective tissue (i.e. codes M00–M99 using the International Classification of Diseases version 10) by Tsimane Health and Life History Project (THLHP) physicians. By age 70 most adults report at least some physical discomfort during activities of daily living (e.g. bathing).

## Study design and participants

The Tsimane thoracic CT sample used to measure thoracic vertebral body BMD and fracture presence and severity includes all individuals who met the inclusion criteria of self-identifying as Tsimane and who were aged 40+ at study enrollment (mean ± SD age = 55.9 ± 9.9, age range: 41–91 years, 48% female, n = 493) (see Appendix for additional details and *Appendix 1—table 1* for descriptives of study variables). This CT sample is thought to be representative of all Tsimane aged 40+ years . Younger adults (<40 years old) were excluded because CT scans were performed as part of a broader project on atherosclerosis, including arterial plaque formation, which is usually negligible at younger ages for physically active adults. No Tsimane were excluded based on any health condition that can affect BMD or fracture risk.

Institutional IRB approval was granted by UNM (HRRC # 07–157) and UCSB (# 3-16-0766), as was informed consent at three levels: (1) Tsimane government that oversees research projects, (2) village leadership, and (3) study participants.

## Thoracic CT

Thoracic CT scans were performed at the Hospital Presidente German Busch in Trinidad, Bolivia using a 16-detector row scanner (GE Brightspeed, Milwaukee, WI, USA). A licensed radiology technician acquired a single non-contrast thoracic scan, which was supervised and reviewed by at least one of the HORUS study team cardiologists and radiologists with whom we collaborate to study cardiovascular functioning (*Kaplan et al., 2017*). Breath-hold instructions were given in the Tsimane language to minimize respiratory motion artifact and misregistration. CT settings were as follows: 250 ms exposure, 2.5 mm slice thickness, 0.5 s rotation speed, 120 kVp, and 40 mA (see *Kaplan et al., 2017*; *Stieglitz et al., 2019* for additional details).

## Thoracic vertebral body BMD

CT-derived vertebral body BMD estimates are highly accurate and strongly positively correlated (Pearson r's > 0.9) with ash weight of dry bone (*Ho et al., 1990*; *Reinbold et al., 1991*). Thoracic vertebral body BMD was measured manually in each of three consecutive vertebrae (T7–T10 range) by a radiologist with 20+ years of experience (see Appendix for additional details). Lumbar vertebrae were not assessed as they did not appear in the CT image field of view. BMD measurement started at the level of the section that contained the left main coronary artery (LMCA) caudally (beginning at either T7 or T8, depending on the origin of the LMCA). The LMCA was set as the reference site to allow reproducible detection of a spinal level; because the LMCA is covered in 100% of images and the field of view can be completely reconstructed, it is an optimal reference point to locate the starting measurement level. The LMCA has previously been found to originate at T7 about as frequently as T8 (*Budoff et al., 2010*).

The region of interest was located around the center of the vertebral body, with the edges 2–3 mm from the vertebral shell. This distance ensured that vertebral body BMD measurements

excluded the cortical bone of the vertebral shell. For each vertebra, the radiologist manually positioned a circular region of interest while demarcating cortical from trabecular bone based on visual inspection. Any area with large vessels, bone island fractures, and calcified herniated disks were excluded as much as possible from the region of interest with use of the manual free tracing protocol. Mean BMD for the three consecutive thoracic vertebrae was then calculated. This CT-derived thoracic vertebral body BMD measure is strongly positively correlated (Pearson r's > 0.9) with CT-derived lumbar vertebral body BMD measures (*Budoff et al., 2012*). CT-derived BMD estimates can be obtained with and without calibration phantoms. Phantomless BMD estimates correlate strongly (Pearson r = 0.99) with standard phantom-based CT BMD estimates (*Budoff et al., 2013*). Hounsfield units were converted to BMD ($mg/cm^3$) using a calibration phantom of known density, or a scanner-specific mean calibration factor for the T7–T10 vertebrae from scans performed without the phantom.

## Thoracic vertebral body fracture

For each participant the radiologist also classified seven vertebrae (T6–T12) according to Genant's semi-quantitative technique (GST) (*Genant et al., 1993*). While there is no consensus regarding the radiologic definition of vertebral fracture, the GST provides highly reproducible diagnosis of fractures and is currently the most widely used clinical technique for identifying and diagnosing fracture (*Shepherd et al., 2015*). Based on visual inspection, each vertebra is rated according to severity of loss of vertebral height and other qualitative features, including alterations in shape and configuration of the vertebra relative to adjacent vertebrae and expected normal appearances. Each vertebra is classified into one of five categories: normal (grade 0); mild fracture (grade 1; approximately 20–25% reduction in anterior, middle, and/or posterior vertebral height, and a 10–20% reduction in projected vertebral area); moderate fracture (grade 2; 25–40% reduction in any height and a 20–40% reduction in area); and severe fracture (grade 3; >40% reduction in any height and area). A grade 0.5 indicates borderline deformed vertebra (<20% reduction in any height) that is not considered to be a definitive fracture (see Appendix for additional details). Each participant is assigned one summary grade measure of all seven vertebrae. Participants with >1 vertebral deformity are classified according to their most severe deformity. Participants are considered to present fracture if any vertebral body is graded at least mildly deformed (i.e. grade ≥1); participants are considered to present no fracture if graded 0 or 0.5. Given lower observer agreement for mild fractures (the most common) relative to moderate and severe fractures (*Lentle et al., 2018*), we repeated some analyses using a more conservative fracture definition (i.e. grade ≥2). Unless otherwise noted we report results using the former definition.

## Subsistence involvement and hunting productivity losses

A few days or weeks before traveling from their home villages to Trinidad for CT scanning, participants received in their villages a medical exam from a Bolivian physician and bilingual (Spanish–Tsimane) physician assistant from the THLHP. During the medical exam, participants were asked whether they continued or completely ceased involvement in routine subsistence tasks (e.g. 'Do you still hunt? Have you hunted in the past few months, or have you stopped hunting altogether?'). Participants were queried about the following four tasks: hunting (men only), tree chopping (men only), weaving (women only), and ability to walk an entire day (both sexes) (also see *Stieglitz et al., 2015b*). Other subsistence tasks (e.g. fishing and collecting fruit) were omitted due to time constraints during data collection. Women were not queried about hunting or tree chopping (nor were men about weaving) because they rarely participate in these tasks (*Gurven et al., 2009*). We were able to minimize assignment of 'false negatives' (i.e. coding capable participant $i$ as ceasing task $j$ due to a temporary illness which only inhibited $i$'s involvement in $j$ in the shorter but not longer term) by cross-checking participant response stability over time using repeated measures from medical exams performed in other project years (before and in a few cases after CT scanning; 98% of CT study participants received at least one medical exam in other [i.e. non-CT scanning] project years [mean ± SD number of medical exams/participant = 4 ± 2, min = 1, max = 8]). Responses were generally stable within individuals over time, resulting in very few false negatives for a given task. If a participant reported ceasing a given task in a given year, the physician then asked whether cessation

resulted from specific functional limitations related to sensory perception (e.g. difficulty seeing or hearing), endurance (e.g. feeling tired or weak), and mobility limitations (e.g. hip problems).

Estimates of Tsimane age-specific daily caloric production (total and task-specific [e.g. hunting and fishing]) and time allocation have been published elsewhere (*Gurven et al., 2012*; *Gurven et al., 2009*; *Hooper, 2011*; *Hooper et al., 2014*; *Hooper et al., 2015*; *Kaplan et al., 2010*; *Schniter et al., 2015*; *Stieglitz et al., 2014*). Briefly, production was estimated by interviewing adults once or twice per week from January 2005 to December 2009 about time allocation and production for each co-resident individual over age 6 during the previous 2 days (n = 1245 individuals from 11 villages). Each family was interviewed an average of 46 times (SD = 20), yielding an average of 93 sample days per individual. Each co-resident individual contributes production data, regardless of their BMD or fracture status (which was unknown at the time of economic data collection); thus, from individual-level production data we cannot determine whether adults with fracture or lower BMD have lower production efficiency (e.g. kcals/hr) than adults without fracture or higher BMD.

To estimate men's hunting productivity losses associated with fracture and lower BMD, we multiply men's average age-specific daily hunting production by the probability that men still hunt at a given age. This probability that men still hunt is modeled using binary logistic regression with the following covariates: age (years), fracture status (yes/no), and BMD (mg/cm$^3$; also see Appendix). This probability thus serves as a multiplier adjusting men's average daily hunting production for disability associated with fracture and lower BMD. To determine productivity loss we take the difference in estimated hunting kcals/day at each age for a given fracture and/or BMD value. For example, if a 40-year-old produces 2000 hunting kcals/day, and if the probability that a 40-year-old without fracture still hunts is 0.99 (vs. 0.85 for a 40-year-old with fracture), then the predicted hunting production is 1980 kcals/day for a 40-year-old without fracture (i.e. 2000 × 0.99), and 1700 kcals/day for a 40-year-old with fracture (i.e. 2000 × 0.85; productivity loss = 280 hunting kcals/day). Similarly, if a 50-year-old produces 1800 hunting kcals/day, and if the probability that a 50-year-old without fracture still hunts is 0.9 (vs. 0.75 for a 50-year-old with fracture), then the predicted hunting production is 1620 kcals/day for a 50-year-old without fracture and 1350 kcals/day for a 50-year-old with fracture (productivity loss = 270 hunting kcals/day; that is, 10 fewer daily lost kcals compared to age 40). We assume that hunting return rates (i.e. kcals/hr hunting) are identical for men regardless of fracture status or BMD. Since we do not assume reduced hunting efficiency for men with fracture or lower BMD, the productivity losses reported here may be conservative, and result entirely from complete hunting cessation. This hunting productivity loss is our estimate of the economic cost associated with compromised bone strength. We estimate this cost in two ways: (1) daily loss at age *x*, as in the prior example of a 40- and 50 year-old; and (2) expected cumulative future losses from age *x* onward, where future losses are discounted by mortality. Conceptually, this latter value is similar to Fisher's reproductive value (expected future reproduction at each age) but substitutes age-specific fertility with age-specific caloric production from hunting, given Tsimane population age structure and men's mortality rates (*Gurven et al., 2012*).

## Socio-demographics and anthropometrics

Birth years were assigned based on a combination of methods described elsewhere (see Appendix for additional details). Height and weight were measured during THLHP medical exams using a Seca portable stadiometer and Tanita scale. The scale uses bioelectrical impedance to estimate body fat percentage using proprietary estimation equations. Using weight and percent body fat we calculated fat mass (weight × percent body fat) and fat-free mass (weight − fat mass).

## Data analysis

In descriptive analyses we report fracture prevalence for each sex and test for effects of age and BMD on fracture risk using log-binomial generalized linear models. Chi-squared tests are used to compare the differences in subsistence participation (e.g. whether continuing or completely ceasing hunting or tree chopping) by fracture status. Binary logistic regression is used to model the probability of subsistence task cessation as a function of fracture status and BMD after adjusting for age. We use stepwise regression with fracture status included first, and then BMD, and we compare models based on Akaike information criterion (AIC). False discovery rate (FDR) q-values were computed

using the R package 'qvalue' to account for the multiple testing burden across different subsistence tasks. Parameter estimates are reported as odds ratios (ORs) or predicted probabilities.

We apply several indirect methods for assessing reverse causality, whereby ceasing task performance leads to compromised bone strength (we cannot determine from the CT scan or any other THLHP data when fracture occurred). First, we use chi-squared tests to compare differences by fracture status in self-reported reasons for subsistence task cessation, including deficits in sensory perception, strength, endurance, and mobility. If fractures cause task cessation, then those with fractures should be more likely to report mobility limitations as a main reason for task cessation, rather than other reasons. Second, we use general linear models to compare BMD of adults who continue vs. cease task participation (adjusting for age and fat-free mass). This comparison helps determine whether adults who cease participation experience greater risk of skeletal fragility overall. Lastly, for reasons described below, we test for a mediating effect of anthropometric status in logistic regressions of task cessation on fracture status and BMD. For all analyses we use $\alpha = 0.05$ as the cutoff for statistical significance (we report both p-values and FDR q-values where relevant). Participants with any missing values are removed from analyses. Data that support the findings of this study are available on Dryad (https://doi.org/10.5061/dryad.h44j0zphj).

## Results

### Descriptives: thoracic vertebral body fracture prevalence and covariation with age and thoracic vertebral body BMD
#### Men
Prevalence of any thoracic vertebral body fracture (i.e. grade $\geq$1; including mild, moderate, or severe) for men aged 40+ years is 36% (95% CI: 30–42) (*Figure 2*). Using a more conservative fracture definition (i.e. grade $\geq$2; including only moderate or severe), men's prevalence is 11% (95% CI: 7–15). Neither men's age nor thoracic vertebral body BMD is significantly associated with fracture risk (*Figure 2*), regardless of fracture grade.

#### Women
Prevalence of any grade 1 fracture for women aged 40+ years is 19% (95% CI: 14–24) (*Figure 3*). Women's grade 2 fracture prevalence is 7% (95% CI: 4–10). Women's age does not significantly predict fracture risk in univariate models, regardless of fracture grade. BMD is inversely associated with women's fracture risk after adjusting for age (grade 1: adjusted relative risk$_{BMD per SD increase}$=0.49, 95% CI: 0.34–0.72, p<0.001; grade 2: adjusted relative risk$_{BMD per SD increase}$=0.25, 95% CI: 0.11–0.57, p=0.001; see *Figure 3*); in both of these models that include BMD as a covariate, age is inversely and significantly associated with fracture risk.

### Hunting cessation is associated with thoracic vertebral body fracture and, albeit weakly, with lower BMD
Thirty-eight percent of men with fracture (grade $\geq$1) ceased hunting, compared to 13% with no fracture (*Table 1*). Results are similar using a more conservative fracture definition (i.e. grade $\geq$2; see *Appendix 1—table 2*; subsequently we only present results where fracture grade $\geq$1). After adjusting for age, odds of hunting cessation are 7.3 times greater (95% CI: 3.3–17.6, p<0.001, FDR q < 0.001, n = 256) for men with vs. without fracture (*Appendix 1—table 3*: Model 1). The association between odds of hunting cessation and fracture increases slightly (adjusted OR$_{Fracture}$ = 7.4) after also including thoracic vertebral body BMD as a covariate (adjusted OR$_{BMD per SD}$=0.62, 95% CI: 0.38–0.99, p=0.054, FDR q = 0.161, *Appendix 1—table 3*: Model 2).

### Hunting cessation associated with thoracic vertebral body fracture and lower BMD generates substantial productivity losses
#### Fracture
*Figure 4* shows men's age-specific hunting productivity by fracture status (see *Appendix 1—table 4* for production values and additional details). Men with fracture under-produce to a lesser extent at younger vs. older ages because of high hunting participation rates by younger men regardless of

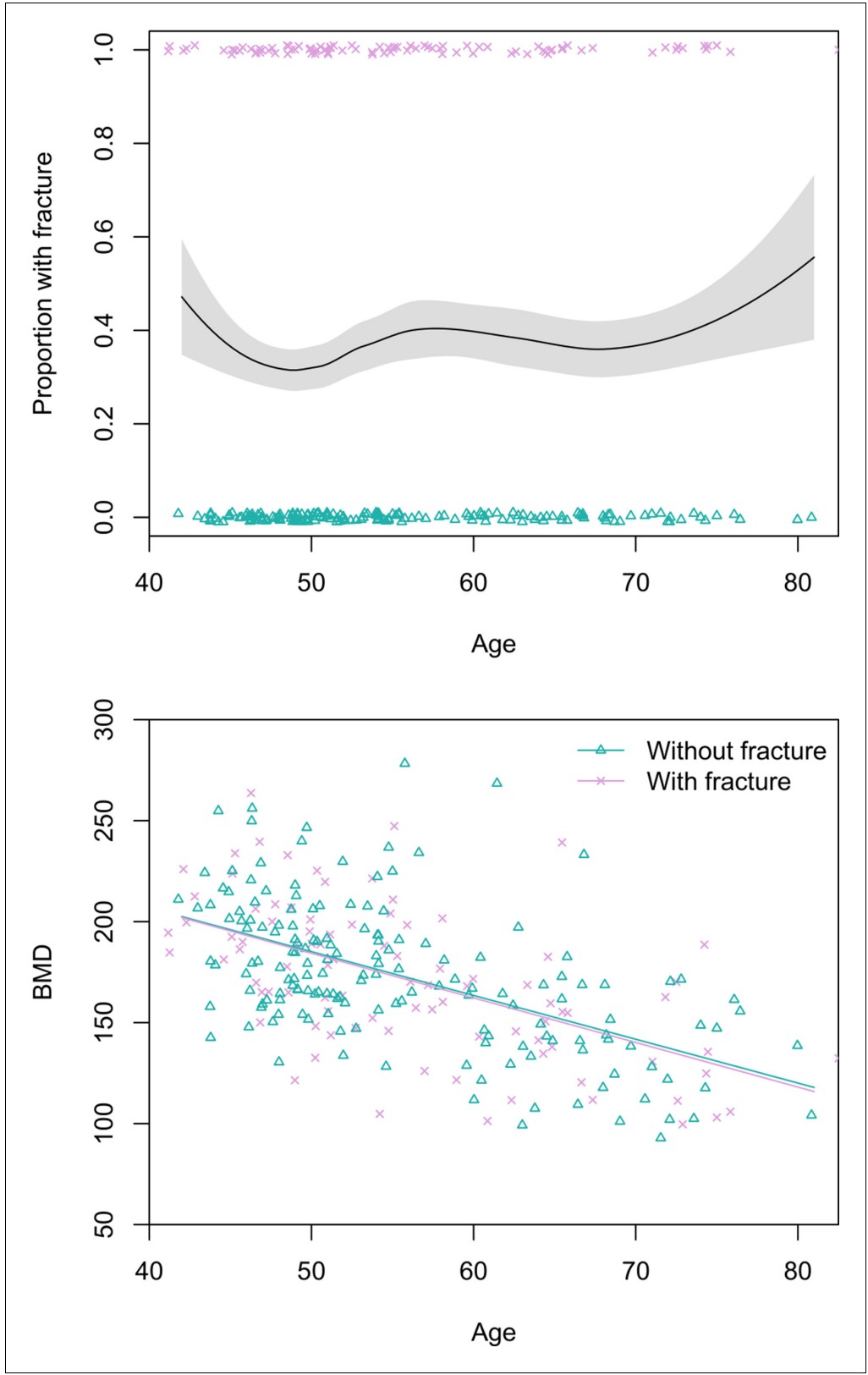

**Figure 2.** Top: Proportion of Tsimane men with thoracic vertebral body fracture (grade ≥1) by age, estimated using the loess function. The shaded region shows ±1 SE, and jittered data points represent fracture status. Bottom: scatterplot of thoracic vertebral body bone mineral density (BMD) by age and fracture status, including linear regression lines for each fracture status. N = 256 men.

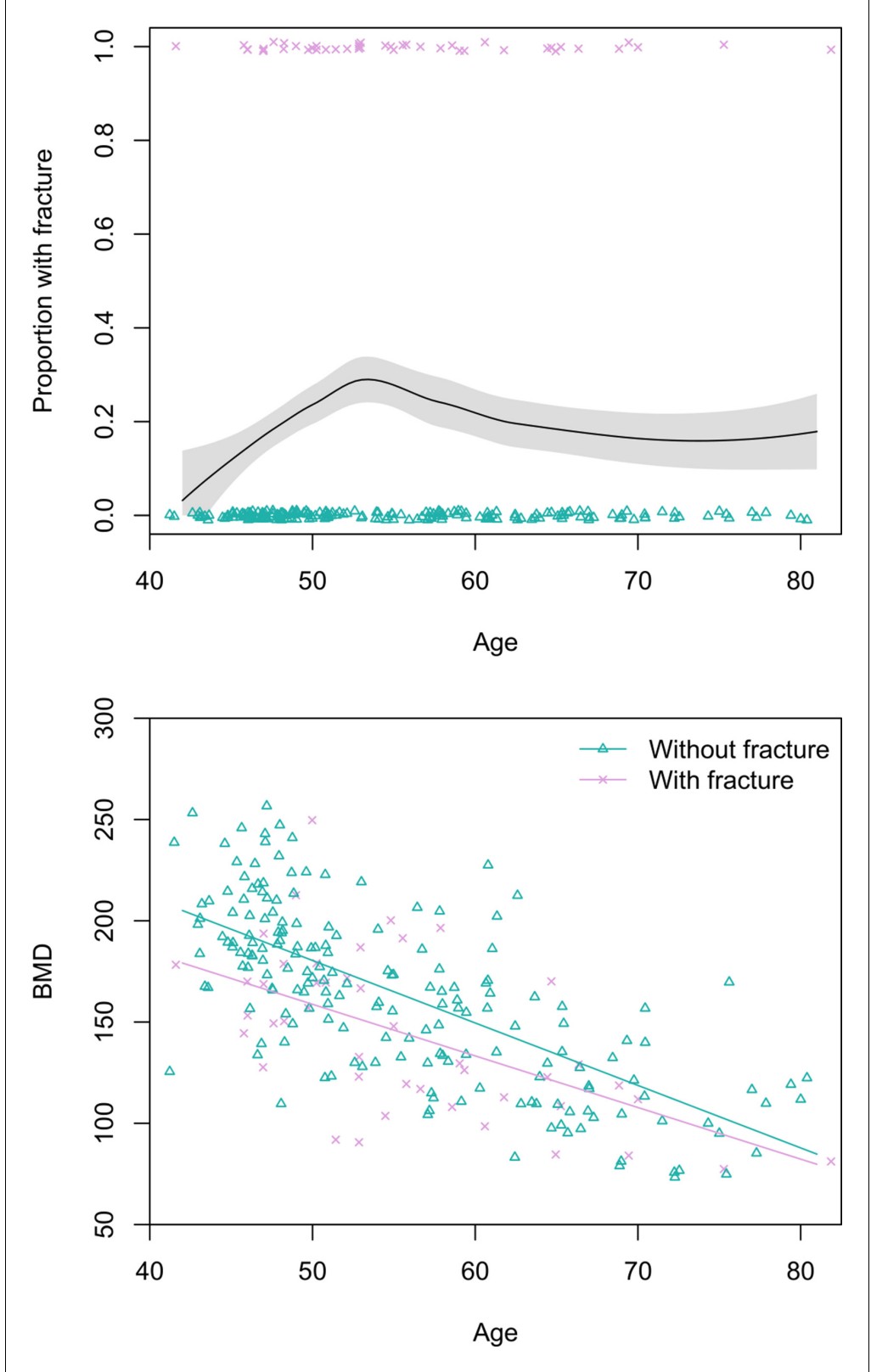

**Figure 3.** Top: Proportion of Tsimane women with thoracic vertebral body fracture (grade ≥1) by age, estimated using the loess function. The shaded region shows ±1 SE, and jittered data points represent fracture status. Bottom: scatterplot of thoracic vertebral body bone mineral density (BMD) by age and fracture status, including linear regression lines for each fracture status. N = 237 women.

**Table 1.** Percentage of men (95% CI) who completely ceased hunting by age and thoracic vertebral body fracture status.

| Age category (years) | % ceased hunting with fracture | N | % ceased hunting without fracture | N |
|---|---|---|---|---|
| 40–49 | 8˘ (<1–19) | 26 | 0 (———) | 55 |
| 50–59 | 33*** (17–50) | 36 | 4 (<1–9) | 56 |
| 60–69 | 50˘ (24–76) | 18 | 27 (12–42) | 37 |
| 70+ | 100* (———) | 12[†] | 63 (36–89) | 16[‡] |
| Total | 38*** (28–48) | 92 | 13 (8–19) | 164 |

˘p≤0.1, *p≤0.05, **p≤0.01, ***p≤0.001 ($\chi^2$ or Fisher's Exact Test [vs. no fracture]).

[†]Max age = 83 years (mean ± SD = 75 ± 4).

[‡]Max age = 81 years (mean ± SD = 74 ± 3).

fracture status (*Table 1* and *Appendix 1—table 2*). As hunting cessation becomes increasingly likely with age, particularly for men with fracture, daily productivity losses associated with fracture (i.e. the hunting productivity differential for men with vs. without fracture) increase until age 61, when men with fracture produce 481 fewer hunting kcals/day than men without fracture (1007 vs. 526 kcals/day, respectively). This loss of 481 kcals/day – the equivalent of 120 g of protein (or 53 g of fat) – represents 19% of the mean daily per-capita energy intake for a Tsimane man that age (2592 kcals), and 52% of the mean intake from protein and fat (919 kcals). By comparison, this loss of 481 hunting kcals/day at age 61 exceeds daily fishing production at that age by 171 kcals (based on men's observed age-specific daily fishing production [see Materials and methods and *Figure 4*]). Daily hunting productivity losses associated with fracture exceed fishing productivity for 14 years, from ages 55 to 68 (by 29–171 kcals/day). In this same age range, based on observed age-specific return rates and time allocation, to wholly offset hunting productivity losses a Tsimane man would have to fish an additional 51–76 min/day (e.g. from an observed 49 to an expected 125 fishing min/day at age 61 [an increase of 155%]; assuming identical fishing return rates regardless of fracture status and BMD; see Materials and methods). As men's hunting participation rates decline with age regardless of fracture status, hunting productivity losses associated with fracture also decline, reaching zero by age 75 (when hunting returns reach zero; *Figure 4*).

*Figure 4* also shows expected cumulative future productivity over the rest of life from a specific age onward, discounted by mortality. Future productivity losses associated with fracture (i.e. the hunting productivity differential for men with vs. without fracture) peak at age 40, when men with fracture can expect to lose 1.15 million kcals over their remaining life; at that age, this loss represents 14% of men's total future hunting production (8.49 million kcals). Future hunting productivity losses associated with fracture exceed future fishing productivity for 16 years, from ages 49 to 64 (by 2–32% [or 4772–96,419 expected cumulative future kcals]; *Figure 4*). In this same age range, based on the maximum observed fishing return rate (420 kcals/hr at age 49) and observed age-specific time allocation (e.g. 53 fishing min/day at age 49), to wholly offset future hunting productivity losses associated with fracture a Tsimane man would have to fish an additional 258–1780 hr (i.e. 324–2012 additional future fishing days). Here we applied the maximum observed return rate to derive a conservative estimate. If instead we apply the minimum rate (360 kcals/hr at age 64), then to wholly offset future hunting productivity losses associated with fracture a man would have to fish an additional 302–2076 hr (i.e. 378–2347 additional future fishing days).

## Lower BMD

We now estimate additional productivity losses (again expressed as the hunting productivity differential) associated with having lower BMD. Losses are greatest for men with fracture and lower BMD vs. men without fracture and higher BMD (*Figure 5*; see *Appendix 1—table 5* for production values and additional details). In this case, peak losses at a given age (age 60: 650 kcals/day – the equivalent of 163 g of protein [or 72 g of fat]) represent 25% of the mean daily per-capita energy intake for a Tsimane man that age and 70% of the mean intake from protein and fat. By comparison, this loss of 650 hunting kcals/day at age 60 exceeds daily fishing production at that age by 333 kcals (based on men's observed age-specific daily fishing production [see Materials and methods]). Hunting productivity losses for men with fracture and lower BMD (vs. men without fracture and higher BMD)

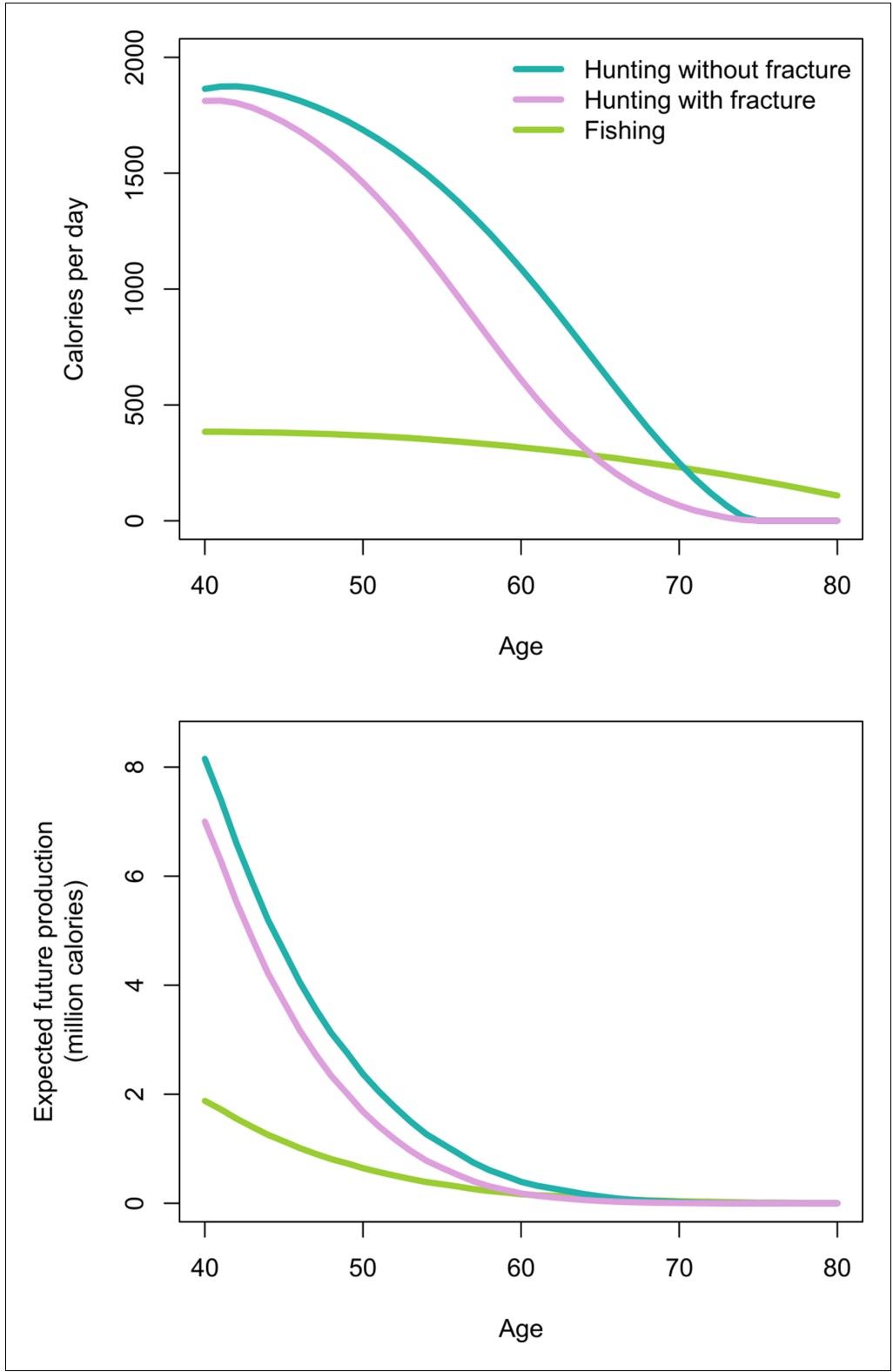

**Figure 4.** Top: Age-specific productivity (kcals/day) for hunting by men's fracture status, and for fishing (all men). Bottom: expected cumulative future productivity (millions of kcals) from age *x* onward, discounted by mortality, for hunting by men's fracture status, and for fishing (all men).

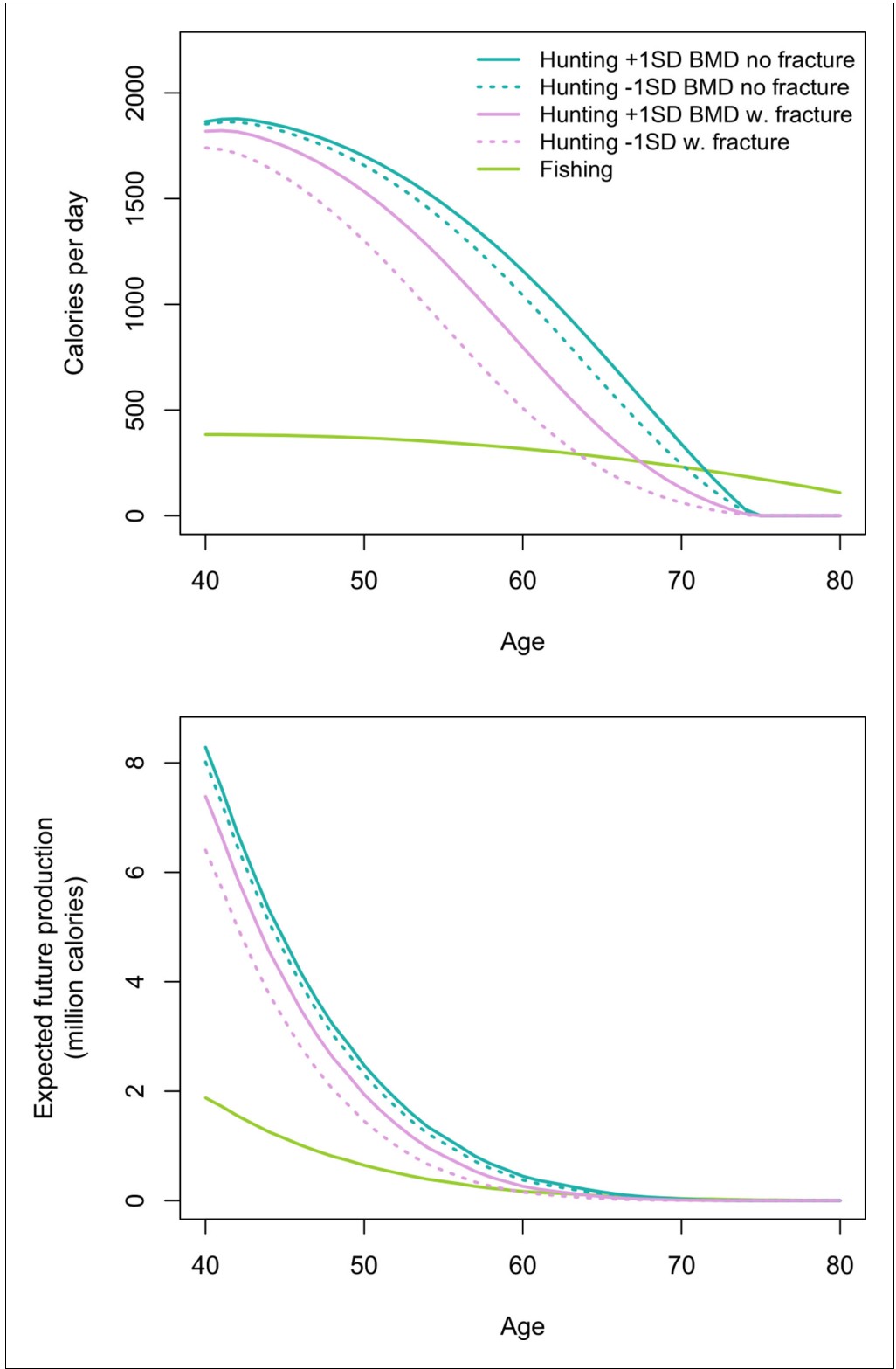

**Figure 5.** Top: Age-specific productivity (kcals/day) for hunting by men's fracture status and bone mineral density (BMD), and for fishing (all men). For illustrative purposes we show daily hunting production for men with +1 SD and −1 SD of the BMD mean. Bottom: expected cumulative future productivity (millions of kcals) from age *x* onward, discounted by mortality, for hunting by men's fracture status and BMD, and for fishing (all men).

exceed fishing productivity for 21 years, from ages 50 to 70 (by 33–335 kcals/day). In this same age range, based on observed age-specific return rates and time allocation, to wholly offset hunting productivity losses a Tsimane man would have to fish an additional 54–102 min/day (e.g. from an observed 49 to an expected 151 fishing min/day at age 60 [an increase of 205%]; assuming identical fishing return rates regardless of fracture status and BMD; see Materials and methods). Hunting productivity losses are smallest for men without fracture and higher vs. lower BMD (max = 136 kcals/day at age 65; *Figure 5*).

Expected cumulative future losses are also greatest for men with fracture and lower BMD vs. men without fracture and higher BMD (*Figure 5*). At the peak (age 40) the former are expected to produce 1.88 million fewer future hunting kcals than the latter. At that age this loss represents 22% of men's total future hunting production. Future hunting productivity losses for men with fracture and lower BMD vs. men without fracture and higher BMD exceed future fishing productivity for 28 years, from ages 40 to 67 (by 0.1–81% [or 1193–383,902 expected cumulative future kcals]). In this same age range, based on the maximum observed fishing return rate and observed age-specific time allocation, to wholly offset future hunting productivity losses a Tsimane man would have to fish an additional 160–4443 hr (i.e. 207–4849 additional future fishing days). Here, again, we applied the maximum observed return rate to derive a conservative estimate. If instead we apply the minimum rate, then to wholly offset future hunting productivity losses a man would have to fish an additional 202–5594 hr (i.e. 261–6105 additional future fishing days).

### Tree chopping cessation is associated with thoracic vertebral body fracture but not lower BMD

Forty-six percent of men with fracture ceased chopping trees, compared to 21% with no fracture (*Table 2*). After adjusting for age, odds of tree chopping cessation are 6.9 times greater (95% CI: 3.1–16.6, p<0.001, FDR q < 0.001, n = 256) for men with vs. without fracture (*Appendix 1—table 6*: Model 1). The association between odds of tree chopping cessation and fracture weakens slightly (adjusted $OR_{Fracture}$ = 6.8) after also including thoracic vertebral body BMD as a covariate, which is not associated with tree chopping cessation (adjusted $OR_{BMD\ per\ SD}$=0.75, 95% CI: 0.47–1.16, p=0.204, FDR q = 0.306, *Appendix 1—table 6*: Model 2).

### Weaving cessation is not associated with thoracic vertebral body fracture but is associated, albeit weakly, with lower BMD

Twenty percent of women with fracture ceased weaving, compared to 13% with no fracture (*Table 3*). After adjusting for age, odds of weaving cessation are not significantly associated with fracture (adjusted $OR_{Fracture}$ = 2.2, 95% CI: 0.8–6.4, p=0.129, FDR q = 0.172, n = 237, *Appendix 1—table 7*: Model 1). Weaving is the only task examined for which fracture is not associated with compromised participation. Adding BMD as a covariate further weakens the association between fracture and weaving cessation (adjusted $OR_{Fracture}$ = 1.8; adjusted $OR_{BMD\ per\ SD}$=0.51, 95% CI: 0.23–1.05, p=0.079, FDR q = 0.158; *Appendix 1—table 7*: Model 2).

Regardless of fracture status or BMD, all women who stopped weaving reported experiencing problems in all bodily regions (i.e. hip, back, and hands) mentioned by the physician during medical exams, and nearly all of these women also reported visual impairment (*Appendix 1—table 8*).

**Table 2.** Percentage of men (95% CI) who completely ceased tree chopping by age and thoracic vertebral body fracture status.

| Age category (years) | % ceased tree chopping with fracture | N | % ceased tree chopping without fracture | N |
|---|---|---|---|---|
| 40–49 | 15** (1–30) | 26 | 0 (———) | 55 |
| 50–59 | 39*** (22–56) | 36 | 4 (<1–9) | 56 |
| 60–69 | 67 (43–91) | 18 | 54 (37–71) | 37 |
| 70+ | 100 (———) | 12[†] | 81 (60–100) | 16[‡] |
| Total | 46*** (35–56) | 92 | 21 (15–28) | 164 |

ˆp≤0.1, *p≤0.05, **p≤0.01, ***p≤0.001 ($\chi^2$ or Fisher's Exact Test [vs. no fracture]).

[†]Max age = 83 years (mean ± SD = 75 ± 4).

[‡]Max age = 81 years (mean ± SD = 74 ± 3).

**Table 3.** Percentage of women (95% CI) who completely ceased weaving by age and thoracic vertebral body fracture status.

| Age category (years) | % ceased weaving with fracture | N | % ceased weaving without fracture | N |
|---|---|---|---|---|
| 40–49 | 0 (———) | 11 | 0 (———) | 72 |
| 50–59 | 14 (<1–29) | 22 | 5 (<1–11) | 60 |
| 60–69 | 33 (<1–72) | 9 | 24 (10–39) | 37 |
| 70+ | 75 (<1–100) | 4[†] | 55 (32–77) | 22[‡] |
| Total | 20 (8–31) | 46 | 13 (8–17) | 191 |

¨p≤0.1, *p≤0.05, **p≤0.01, ***p≤0.001 ($\chi^2$ or Fisher's Exact Test [vs. no fracture]).

[†]Max age = 91 years (mean ± SD = 80 ± 9).

[‡]Max age = 91 years (mean ± SD = 77 ± 6).

## Restricted day range is associated with thoracic vertebral body fracture and, albeit weakly, with lower BMD

Sixty-nine percent of adults with fracture are unable to walk a full day, compared to 38% with no fracture (*Table 4*). After adjusting for age and sex, odds of inability to walk all day are 8.2 times greater (95% CI: 4.8–14.5, p<0.001, FDR q < 0.001, n = 493) for adults with vs. without fracture (*Appendix 1—table 9*: Model 1). The association between odds of inability to walk all day and fracture weakens slightly (adjusted $OR_{Fracture}$ = 7.8) after also including thoracic vertebral body BMD as a covariate (adjusted $OR_{BMD\ per\ SD}$=0.77, 95% CI: 0.57–1.03, p=0.082, FDR q = 0.109, *Appendix 1—table 9*: Model 2).

## Suggestive evidence of fracture preceding task cessation, rather than the reverse

Men with fracture are more likely than men without fracture to report mobility limitations due to hip problems as a reason for hunting and tree chopping cessation (*Appendix 1—tables 10* and *11*). This is noteworthy because among men who ceased hunting and tree chopping, men with fracture are younger, on average, than men without fracture, and all else equal one expects younger men to be *less* likely than older men to report hip problems (for men ceasing hunting: mean age by fracture status = 63 vs. 68; for men ceasing tree chopping: mean age = 62 vs. 68). Consistent with their younger age, men with fracture are less likely than men without fracture, albeit not significantly, to report problems with endurance (i.e. feeling weak and tiring easily) or sensory processing (i.e. auditory and visual) as reasons for hunting cessation (*Appendix 1—table 10*). It is thus less likely that men with fractures are in poorer condition more generally, and more likely that fractures contributed to the mobility complaints that led men to stop hunting and tree chopping. As with hunting and tree chopping, among adults who are unable to walk all day, those with fracture are younger than those without fracture (mean age = 59 vs. 62) (there are no differences by fracture status in self-reported reasons for inability to walk all day [see *Appendix 1—table 12*]).

After adjusting for age and fat-free mass, men who are no longer able to hunt, chop trees, and walk all day do not differ in their BMD vs. men who still hunt (95% CI: −18.1 to 4.4 mg/cm$^3$), chop

**Table 4.** Percentage of adults (95% CI) who are unable to walk a full day by age and thoracic vertebral body fracture status.

| Age category (years) | % unable to walk all day with fracture | N | % unable to walk all day without fracture | N |
|---|---|---|---|---|
| 40–49 | 49*** (32–66) | 37 | 19 (12–26) | 127 |
| 50–59 | 64*** (51–77) | 58 | 27 (19–35) | 116 |
| 60–69 | 89** (76–100) | 27 | 64 (52–75) | 74 |
| 70+ | 100¨ (———) | 16[†] | 84 (72–96) | 38[‡] |
| Total | 69*** (61–77) | 138 | 38 (33–43) | 355 |

¨p≤0.1, *p≤0.05, **p≤0.01, ***p≤0.001 ($\chi^2$ or Fisher's Exact Test [vs. no fracture]).

[†]Max age = 91 years (mean ± SD = 76 ± 6).

[‡]Max age = 91 years (mean ± SD = 76 ± 5).

trees (95% CI: −15.0 to 7.1 mg/cm$^3$), and can walk all day (95% CI: −10.2 to 7.9 mg/cm$^3$). This suggests that men who cease these activities do not experience greater risk of overall skeletal fragility. For women, inability to walk all day is associated with reduced BMD (95% CI: −25.3 to −6.1 mg/cm$^3$) after adjusting for age and fat-free mass. Lastly, if hunting or tree chopping cessation preceded fracture (e.g. because reduced meat consumption weakened bone structural integrity, or because muscle degeneration from task cessation reduced bone loading and structural integrity), then one might expect anthropometric status to mediate the association between hunting or tree chopping cessation and fracture. Yet these associations strengthen after adjusting for anthropometrics (i.e. weight, fat-free mass, and fat mass). Anthropometric status also does not mediate the association between inability to walk all day and fracture. Together, these results provide suggestive evidence that fracture preceded task cessation, rather than the reverse.

## Discussion

We find among adult Tsimane forager-horticulturalists that compromised bone strength – as indicated by thoracic vertebral body fracture and lower thoracic vertebral body BMD – is associated with diminished subsistence involvement. Compromised bone strength is associated with diminished participation in diverse habitual subsistence tasks, particularly those requiring high levels of strength or endurance (i.e. hunting, tree chopping, and walking long distances). In contrast, participation in less physically demanding but skill-intensive tasks (i.e. tool manufacture/repair) is only modestly associated with bone strength. Since strength and endurance are required to exploit the suite of resources upon which omnivorous hominins rely, functional disability may hinder resource production in costly ways for a comparatively long-living, extractive foraging primate. This inference is supported by prior studies in other contemporary small-scale populations that highlight illnesses or injuries of shorter duration (e.g. acute infection, laceration, and animal bite) as constraints on productivity (*Bailey, 1991*; *Sugiyama and Chacon, 2000*; *Sugiyama, 2004*). As most activities of daily living generate loads on human vertebrae (*Myers and Wilson, 1997*; *Rohlmannt et al., 2001*; *Stewart and Hall, 2006*), and since for many activities the greatest compressive loads along the spine are in the thoracolumbar region (*Bruno et al., 2017*), thoracic vertebral body fracture in particular may be debilitating for an obligate biped even after behavioral modifications that minimize its burden.

Ample bioarchaeological evidence of partially or fully healed fractures (e.g. *Lovejoy and Heiple, 1981*; *Pfeiffer, 2012*), usually from excessive trauma, and developmental abnormalities (*Cowgill et al., 2015*; *Trinkaus, 2018*) suggest in past populations longer-term survival with varying levels of disability. Fragility fractures resulting from reduced bone strength are relatively rare in bio-archaeological studies (*Agarwal, 2008*; *Agarwal and Grynpas, 1996*), even among the elderly, leading some to conclude that osteoporosis was rare or absent in past populations (but see *Curate et al., 2010*; *Dequeker et al., 1997*). This and the fact that age-specific osteoporotic fracture incidence rates appear to be increasing over time in Western populations (*Cooper et al., 2011*) lends support to the notion that osteoporosis is a 'disease of modernity' (cf. *Kralick and Zemel, 2020*; *Lieberman, 2013*). However, few if any bio-archaeologists systematically study thoracic vertebral compression fractures, and a paucity of observed fragility fractures in the archaeological record is partly due to biased preservation of skeletal remains. Accelerated bone loss with age appears to be a basic feature of human aging in past and present populations (*Aspray et al., 1996*; *Kneissel et al., 1997*; *Mays et al., 1998*; *Mays, 1996*; *Wallace et al., 2014*) and has also been documented among free-ranging chimpanzees (*Gunji et al., 2003*; *Morbeck and Galloway, 2002*; *Sumner et al., 1989*). It would thus be premature to conclude that fragility fractures were rare or of minimal consequence throughout human evolution.

We find suggestive evidence that Tsimane fracture precedes subsistence task cessation, rather than the reverse. But there are several possible explanations for the empirical associations reported in this paper (see *Figure 6*) and given the study design we cannot definitively parse among them. Associations between reduced thoracic vertebral body BMD and task cessation may reflect effects on functional ability of other properties of senescence that were not examined here. The present study does not consider diverse mechanisms (e.g. spinal osteoarthritis, restricted range of motion for a given anatomical region) and future research is needed to establish causal relationships between somatic condition, functional ability, and productivity. A candidate model that is only partially evaluated in this paper (*Figure 6*) proposes that senescence induces anatomical and sensory

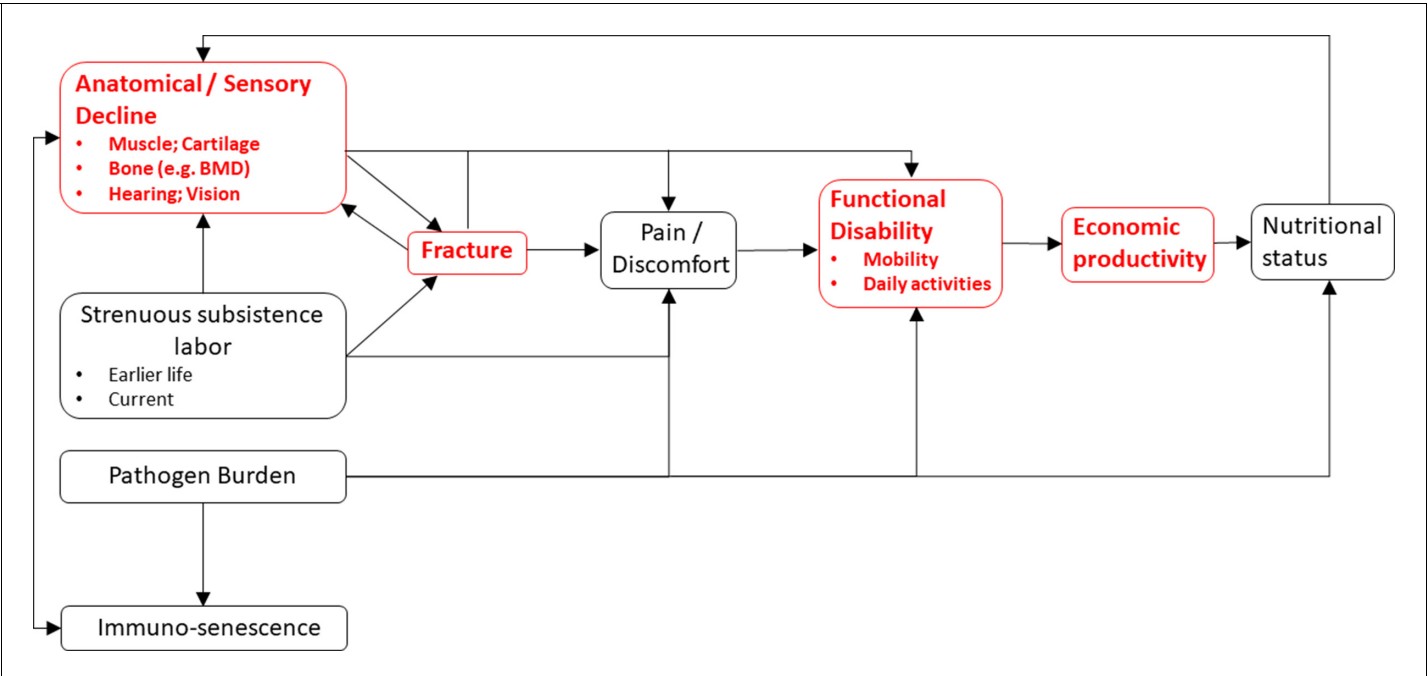

**Figure 6.** A conceptual model linking somatic condition to economic productivity during aging in small-scale subsistence societies. Variables in red are a focus of the present study.

declines – including loss of strength in muscles, bones, and connective tissues – which jointly increase risk of fracture, body pain or discomfort, and functional disability. Senescence-induced declines can also induce pain and disability in the absence of fracture or reduced BMD (e.g. from osteoarthritis). The magnitude and pace of anatomical and sensory decline may be influenced not just by senescence but also by habitual participation in strenuous subsistence labor (i.e. 'mechanical wear and tear'), which can directly cause fracture (e.g. via excessive mechanical stress) and body pain. The manifestation of pain and functional disability following fracture appears to be quite variable both across individuals and within individuals over time (e.g. *Begerow et al., 1999*; *Nevitt et al., 1998*; *O'Neill et al., 2004*; *Suzuki et al., 2008*) and samples are usually not representative of the population since many fractures remain unobserved. Back pain can arise directly from vertebral fracture or indirectly from consequences of spinal deformity or degenerative changes involving paravertebral muscles, intervertebral discs, associated ligaments, and facet joints. Fractures in particular can entail longer-term productivity constraints as they provoke systemic bone losses, incomplete recovery of bone quality (especially for older adults), increased risk of future fractures, and longer-term disability (*Osipov et al., 2018*). Future research is needed to test the empirical adequacy of the hypothesized, sometimes bidirectional relationships shown in *Figure 6*. It is also possible that our results reflect influences of earlier life factors (e.g. activity levels, peak bone mass, and economic productivity) that we do not examine in the present study.

Tsimane thoracic vertebral fractures appear to result from both trauma and senescence (*Stieglitz et al., 2019*) and are not just prevalent among the oldest adults (*Figures 2* and *3*). Tsimane can thus experience a significant portion of adult life with fracture. Tsimane women's thoracic vertebral body BMD is inversely associated with thoracic vertebral body fracture risk, suggesting a role of compromised bone strength in precipitating fracture. For men, we suspect that work-related trauma plays a role in precipitating fracture due to a weaker association between BMD and fracture risk (*Stieglitz et al., 2019*), and our anecdotal observations of high levels of mechanical stress on men's vertebrae from frequent heavy load carrying (e.g. of animal carcasses, and timber for constructing houses). This of course does not preclude a contributing role of compromised bone strength in precipitating men's fracture. Likewise, for women, trauma may also precipitate fracture; some of the greatest compressive vertebral loads occur when weights are carried in front of the body (*Bruno et al., 2017*; *Rohlmannt et al., 2001*), which is how Tsimane women

routinely carry infants and toddlers. High reproductive effort (i.e. early onset of reproduction, on-demand breastfeeding, short interbirth intervals, and high parity) is associated with reduced bone strength for Tsimane women (*Stieglitz et al., 2015a*; *Stieglitz et al., 2019*) and may thus constrain the ability of bone tissue to respond to mechanical loading from a physically active lifestyle. Interpersonal violence likely does not precipitate much fracture among men because such violence is relatively rare and generally does not result in severe injuries when it does occur. However, intimate partner violence against women is prevalent (*Stieglitz et al., 2018*), particularly among younger women, and may precipitate traumatic fracture at diverse skeletal sites.

To more precisely identify the economic cost of disability, we estimated age-specific energetic deficits from hunting cessation by men with compromised bone strength. An advantage of our approach to estimating productivity loss is that it is not subject to recall inaccuracy or subjectivity regarding the meaning of foregone production. Energetic deficits are substantial, averaging from ages 40 to 74 years 261 lost hunting kcals/day for men with vs. without fracture (*Figure 4* and *Appendix 1—table 4*). Average production deficits are even greater for men with both fracture and lower BMD (vs. men without fracture and higher BMD): 397 lost hunting kcals/day, equivalent to 15% of the mean daily per-capita energy intake, and 44% of the mean intake from protein and fat (*Kraft et al., 2018*; *Figure 5* and *Appendix 1—table 5*). By crude comparison, adults in Western industrialized settings with musculoskeletal problems show productivity losses of 10–15% for frequent computer users across various industries (*Hagberg et al., 2002*) and up to 28% for construction workers with more physically demanding jobs (*Meerding et al., 2005*). For osteoporotic adults aged 50+ years in the Netherlands, the cost of clinical spine fracture in terms of work absenteeism exceeds 25% of the mean annual income (*Eekman et al., 2014*). Tsimane age-specific energetic deficits peak in the early 60s (*Figures 4* and *5* and *Appendix 1—tables 4* and *5*), which is near the age at which Tsimane adults achieve peak numbers of dependent offspring and grand offspring (*Gurven and Kaplan, 2007*). The magnitude of these disability costs and their timing in the life cycle may thus pose for Tsimane significant challenges to meeting adult survival needs and/or providing assistance (material or other) to descendant kin. These challenges should vary in severity across populations depending on fertility schedules and dependency loads. Our results suggest that the scope of behavioral responses to compensate for productivity loss is limited to activities with lower strength and endurance requirements. Given the ubiquity across subsistence populations of the sexual division of labor, with men typically focusing on high strength tasks, aging women may maintain peak or near-peak subsistence involvement with relatively minimal disability-related productivity losses, despite increasing frailty. Relatedly, in industrialized populations there is some evidence from population-based cohorts that severe thoracolumbar vertebral deformity is more strongly associated with functional impairments for men than women (*Burger et al., 1997*), perhaps because men load their spines more heavily.

Faced with protein and fat deficits from hunting cessation, one might expect adults with compromised bone strength to increase time spent in other, less physically demanding food acquisition tasks that reliably yield substitutable macronutrients. Fishing is the other major source of protein and fat in small-scale Amazonian subsistence populations, most of whom acquire protein more efficiently from animal- rather than plant-based foods (*Hames, 1989*). Prior behavioral studies of other contemporary Amazonian populations (*Gurven and Kaplan, 2006*) indicate a shift with age in men's time allocation away from hunting and instead toward fishing; for both sexes time spent in fishing increases monotonically throughout adulthood. Fishing is also an important source of protein and fat for Tsimane (*Gurven et al., 2013*; *Hooper, 2011*). We find that, to wholly offset their own hunting productivity losses, men with fracture and lower BMD would have to considerably increase their time spent fishing. A perceived need by adults to offset their own current and/or future energetic deficits (*Figures 4* and *5*) could influence residential decisions by increasing the value of maintaining proximity to fishing locations (e.g. rivers and lakes), particularly during periods of greater fishing efficiency (e.g. dry season). Disability-related production deficits could influence various economic decisions (e.g. by influencing decisions on where to forage on a given day, or where to farm). Minimizing travel costs during food acquisition may be important in light of restricted day ranges among adults with compromised bone strength (see Results). These economic decisions also depend on local demographics and social networks, as productivity losses can be mitigated with the availability of substitute workers. If, for instance, able-bodied adults within the multi-generational sharing unit can

substitute for impaired adults in hunting and meat sharing, then impaired adults may take fewer risks to acquire protein and fat compared to those without buffered networks.

Our estimates of the ways in which Tsimane with compromised bone strength may compensate for hunting productivity loss are crude for several reasons. They do not, for example, consider the possibility of adults being provisioned by others, or pursuing other means of protein and fat acquisition beyond fishing. Tsimane can pursue other low-intensity subsistence tasks (e.g. harvesting cultigens), thereby offering additional opportunities for adults to produce and transfer resources outside of a strict hunting and gathering economy. Our crude estimates nevertheless imply that productivity losses entail behavioral modifications and might affect social organization more broadly. The multigenerational production and sharing units that structure human sociality (*Bird et al., 2019*; *Isaac, 1978*; *Migliano et al., 2017*; *Washburn and Lancaster, 1968*) incorporate divisions of labor whereby impaired adults can still make valuable contributions beyond food acquisition (e.g. through conflict mediation, pedagogy, and leadership). Older adults play an important fitness-enhancing role not just by providing caloric surpluses and transfers, but also by providing childcare, skills and knowledge, and other resources. Tsimane cognitive function is generally well preserved throughout adulthood among both schooled and non-schooled adults (*Gurven et al., 2017a*), consistent with a late-life service niche emphasizing information transmission of accumulated wisdom, experience, and judgment. Aging Tsimane adults, despite their declining physical abilities, are regularly consulted and respected for their knowledge of the natural and spiritual world (e.g. profitable hunting locations; proper ethics), family lineage histories and past events, healing sick group members, making household crafts, and traditional stories and music (*Schniter et al., 2015*; *Schniter et al., 2018*). Aging adults accommodate their changing roles within the family and society and gradually compensate for their increasing frailty by providing valuable expertise across diverse skill- and knowledge-intensive domains that facilitate achieving the childrearing and subsistence goals of the residential group (cf. *Levitin, 2020*).

The extent to which our results generalize across time and space depends on various considerations including age profiles of production, rates of mortality and senescence, and subsistence ecology (e.g. reliance on hunted versus other foods). Fertility patterns are not explicitly considered in the present study, but they can influence both somatic condition, particularly for women (*Jasienska, 2020*; *Ryan et al., 2018*; *Stieglitz et al., 2015a*; *Stieglitz et al., 2019*; *Ziomkiewicz et al., 2016*), and economic productivity (*Hooper et al., 2015*; *Kramer, 2005*). Extensive longevity may be a novel life history feature of modern *H. sapiens* that was absent among prior hominin species; extant hunter-gatherers with relatively minimal acculturation exhibit a modal adult lifespan of 68–78 years (*Gurven and Kaplan, 2007*). It is thus possible that our estimates of disability-related productivity loss are more relevant for understanding lifeways of past hominin populations with more similar life histories as modern *H. sapiens* and perhaps some archaic humans (e.g. *H. heidelbergensis*), but less applicable to Lower Paleolithic and earlier hominins with shorter adult lifespans. Whether lifespans were too short for some fossil hominins to expect significant post-reproductive longevity is vigorously debated (see *Gurven and Kaplan, 2007*; *Hawkes, 2003*; *Hill et al., 2007* and references therein). One study using dental-wear seriation and relative macro-age categories (ratio of old to young) demonstrated an increase in the relative presence of older adults from Australopithecines to early *Homo* and Early Upper Paleolithic humans (*Caspari and Lee, 2004*; but see *Hawkes and O'Connell, 2005*; *Minichillo, 2005*). Re-estimation of several paleo-mortality curves based on hazard analysis and maximum likelihood methods shows a life course pattern more similar to that of modern human hunter-gatherers than previous methods (*Konigsberg and Herrmann, 2006*). Development of reliable paleo-demographic reconstructions of hominin life histories (e.g. see *DeWitte and Stojanowski, 2015*) are critical for determining generalizability of the results presented here. Moreover, disability-related productivity losses could change based on variation in ancestral hominin vertebral morphology that can affect interpretation of fracture results using Genant's semi-quantitative technique (*Trinkaus, 2018*).

## Conclusion

In comparative cross-species perspective, modern human life histories are characterized by delayed peak foraging efficiency (e.g. *Kaplan, 1994*; *Kaplan, 1997*; *Koster et al., 2020*; *Walker et al., 2002*), complex cooperative strategies to produce and rear altricial, slow-growing offspring (e.g. *Hawkes, 2003*; *Kaplan et al., 1985*; *Richerson and Boyd, 2008*; *Wood and Marlowe, 2013*), low

adult mortality rate (e.g. *Hill et al., 2001*; *Wood et al., 2017*), and long post-reproductive lifespan (e.g. *Gurven and Kaplan, 2007*; *Hawkes et al., 1998*; *Kaplan et al., 2010*). Resource transfers from older to younger generations help mitigate risk of production deficits in a skill- and strength-based extractive foraging niche (*Gurven et al., 2012*; *Hawkes et al., 1998*; *Hooper et al., 2015*; *Kaplan et al., 2010*). In this paper we considered energetic deficits associated with compromised bone strength, which may have been a source of disability and constrained economic productivity and resource transfers for extended periods throughout much of human history (*Bailey, 1991*; *Gurven et al., 2000*; *Lambert and Welker, 2017*; *Sugiyama, 2004*). While we provide suggestive evidence of a causal relationship between bone properties and functional disability, changes in bone strength may reflect a combination of earlier life factors (e.g. trauma and activity levels) or other senescent processes.

Perhaps not surprisingly, Ache hunters cite fear of disability as an important reason why they willingly conform to band-wide conventions of need-based food sharing (*Hill and Hurtado, 2009*). It has been proposed that individuals may have survived periods of prolonged disability only via costly provisioning and care by others (*Gurven et al., 2000*; *Sugiyama and Chacon, 2000*), although the extent to which others adjust their activity budgets, food production, and sharing to compensate for disability-related productivity loss is beyond the scope of the present study. Our findings nevertheless suggest that disability is economically costly.

Vertebral deformities are observed in diverse hominin skeletal samples yet the social and behavioral implications of these deformities remain obscure. In the absence of direct evidence of disability-related productivity loss in the fossil record, bone properties may represent a potential proxy for inferring behavioral consequences of skeletal deformity in prior hominin populations. Quantification of microstructural variation in the appendicular and axial skeleton of fossil hominins may provide evidence for inferring foraging behavior, as detailed in the present study. While our ability to make inferences about the past using data collected in contemporary populations is limited, studies of contemporary small-scale subsistence populations like the Tsimane can offer a unique opportunity to examine potential consequences of skeletal deformity in a population-representative adult sample. The present study provides a framework for interpreting morphological signals identified in the fossil record; while this study is narrowly focused on specific morphological features, and while it overlooks important subsistence tasks performed by women, it can nonetheless be an exemplar for future ethno-archaeological inquiry into potential economic and social consequences of disability in past human populations.

## Acknowledgements

We thank the Tsimane for participating and THLHP personnel for collecting and coding data. We also thank the HORUS Study Team for assistance with CT data collection. Amélie Beaudet, George Perry, Susan Pfeiffer, Ian Wallace, and two anonymous reviewers provided useful comments that improved the quality of the manuscript. Data on bone properties used in this study were generated under the supervision of Matthew Budoff. Funding was provided by the National Institutes of Health/National Institute on Aging (R01AG024119), National Science Foundation (1748282), the Center for Evolution and Medicine at Arizona State University, and the University of California-Santa Barbara Academic Senate. JS acknowledges IAST funding from the French National Research Agency (ANR) under the Investments for the Future (Investissements d'Avenir) program, grant ANR-17-EURE-0010. Funding sources had no role in research conduct, study design, or article preparation.

## Additional information

### Funding

| Funder | Grant reference number | Author |
| --- | --- | --- |
| National Institutes of Health | R01AG024119 | Jonathan Stieglitz<br>Benjamin C Trumble<br>Hillard Kaplan<br>Michael D Gurven |

| National Science Foundation | 1748282 | Jonathan Stieglitz |
| Arizona State University | | Benjamin C Trumble |
| University of California, Santa Barbara | | Michael D Gurven |
| Agence Nationale de la Recherche | ANR-17-EURE-0010 | Jonathan Stieglitz |

The funders had no role in study design, data collection and interpretation, or the decision to submit the work for publication.

### Author contributions

Jonathan Stieglitz, Conceptualization, Data curation, Formal analysis, Funding acquisition, Investigation, Visualization, Methodology, Writing - original draft, Project administration, Writing - review and editing; Paul L Hooper, Data curation, Visualization; Benjamin C Trumble, Supervision, Funding acquisition, Project administration, Writing - review and editing; Hillard Kaplan, Supervision, Funding acquisition, Investigation, Project administration; Michael D Gurven, Supervision, Funding acquisition, Investigation, Methodology, Project administration, Writing - review and editing

### Author ORCIDs

Jonathan Stieglitz https://orcid.org/0000-0001-5985-9643
Michael D Gurven http://orcid.org/0000-0002-5661-527X

### Ethics

Human subjects: Institutional IRB approval was granted by UNM (HRRC # 07-157) and UCSB (# 3-16-0766), as was informed consent at three levels: (1) Tsimane government that oversees research projects, (2) village leadership, and (3) study participants.

### Decision letter and Author response

Decision letter https://doi.org/10.7554/eLife.62883.sa1
Author response https://doi.org/10.7554/eLife.62883.sa2

# Additional files

### Supplementary files

• Transparent reporting form

### Data availability

Data that support the findings of this study are available on Dryad.

The following dataset was generated:

| Author(s) | Year | Dataset title | Dataset URL | Database and Identifier |
| --- | --- | --- | --- | --- |
| Stieglitz J | 2020 | Productivity loss associated with physical impairment in a contemporary small-scale subsistence population | https://doi.org/10.5061/dryad.h44j0zphj | Dryad Digital Repository, 10.5061/dryad.h44j0zphj |

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

# Appendix

## Methods

### Study design and participants

Since 2002 the Tsimane (pop. ~16,000) have participated in the ongoing Tsimane Health and Life History Project (THLHP; see *Gurven et al., 2017b* for an overview). All Tsimane residing in study villages are eligible to participate and most participate at least once. THLHP physicians have conducted annual medical exams on Tsimane of all ages since 2002. A mobile team of physicians and Tsimane research assistants collects data in villages on medical histories, functional ability, and aspects of lifestyle (e.g. food production and sharing).

Between July 2014 and September 2015, men and women aged 40+ years from 59 Tsimane villages were invited to participate in the CT scanning project. At the time, there were 1214 eligible adults living in these villages (the only eligibility criteria were being 40+ years old, self-identifying as Tsimane, and willing to participate). A total of 731 adults were present in their villages at the time and subsequently received a CT scan. Transporting participants from their village to the nearby market town of San Borja was logistically complicated (requiring trekking through the forest, dug-out canoes, rafts propelled by poles pushed off the river bottom, trucks, and cars) and can require up to 2 days of travel each way. From San Borja to the Beni department capital of Trinidad (where the hospital containing the CT scanner is located) is an additional 6 hr car ride. Due to these logistical complications, participants not in their village at the time we arrived were not sampled. The Tsimane are semi-mobile and often build secondary houses deep in the forest near their horticultural plots, not returning to their village for extended periods of time. Hunting and fishing trips can last days or weeks, and some men engage in wage labor in San Borja or elsewhere (e.g. rural cattle ranches). In an average village, approximately one-third of individuals are away hunting, fishing, working in their horticultural plots, or in San Borja at any given time. Additionally, a major flood in February 2014 resulted in mass migration from some villages, and the creation of several new villages that were not sampled as part of this study, further reducing the number of individuals that could be sampled in this study. To address potential sources of sample bias, analyses comparing Tsimane who received CTs and those who did not but who participated in the THLHP's medical exams in Tsimane villages were conducted. There were no significant differences in sex, blood pressure, or body fat (see *Kaplan et al., 2017* for these and additional participant details) and thus the CT sample analyzed here is thought to be representative of all Tsimane aged 40+ years.

Of the 731 Tsimane who received a CT scan, CT data from 506 (69%) were selected with no particular criteria to estimate thoracic vertebral body bone mineral density (BMD) and fracture prevalence and severity (radiologist time constraints precluded analysis of all 731 adults). Of the 506 Tsimane contributing BMD and fracture data, 14 (3%) were missing data on subsistence involvement (e.g. whether still able to hunt or walk all day) because of absent or sick THLHP personnel who were unable to collect data, resulting in a final sample of 493 adults. Among these 493 adults, some anthropometric data were missing because of missing or broken equipment (see *Appendix 1—table 1* for sample sizes and descriptives for study variables).

**Appendix 1—table 1.** Descriptives for study variables*.

| Variable | N | Mean | SD | Min | Max |
|---|---|---|---|---|---|
| Thoracic vertebral body BMD (mg/cm$^3$) | 493 | 165.5 | 41.3 | 68.9 | 315.0 |
| Any thoracic vertebral body (T6–T12) fracture (proportion grade $\geq$1) | 493 | 0.28 | 0.45 | 0 | 1 |
| Any thoracic vertebral body (T6–T12) fracture (proportion grade $\geq$2) | 493 | 0.09 | 0.29 | 0 | 1 |
| Age (years) | 493 | 55.9 | 9.9 | 41.2 | 91.0 |
| Sex (proportion male) | 493 | 0.52 | 0.5 | 0 | 1 |
| Height (cm) | 489 | 156.1 | 7.6 | 136.0 | 176.3 |
| Weight (kg) | 489 | 58.7 | 9.8 | 34.6 | 96.9 |

*Continued on next page*

*Appendix 1—table 1 continued*

| Variable | N | Mean | SD | Min | Max |
|---|---|---|---|---|---|
| Body fat (%) | 445 | 21.5 | 8.0 | 5.0 | 46.7 |
| Fat mass (kg) | 445 | 12.9 | 6.1 | 1.9 | 42.1 |
| Fat-free mass (kg) | 445 | 45.8 | 7.9 | 27.8 | 73.1 |

[*]Data were missing for various reasons (see Appendix for details).

## Thoracic vertebral body BMD

CT-measured vertebral body BMD is increasingly used for osteoporosis screening in vivo because of its ability to provide three-dimensional information compared to traditional dual X-ray absorptiometry two-dimensional images.

Intra-observer variability in BMD measurements using the same measurement technique in a different sample (see *Budoff et al., 2010*) was tested on 120 scans by one radiologist, with 1 week intervals between the two readings. To measure inter-observer variability on 67 randomly selected scans, the results obtained by two radiologists who were blinded to all clinical information were compared using this other sample. Intra- and inter-observer variations in BMD measurements were 2.5% and 2.6%, respectively.

## Thoracic vertebral body fracture

Genant's semi-quantitative technique does not distinguish between wedge (i.e. reduced anterior height), biconcave (i.e. reduced central height), or crush (i.e. reduced posterior height) fractures; most fractures contain a combination of these features and are influenced by the local biomechanics of the spinal level involved. Vertebral body fractures are differentiated from other, non-fracture deformities (e.g. osteoarthritis), although these other deformities were not systematically coded.

## Socio-demographics and anthropometrics

Birth years were assigned based on a combination of methods described in detail elsewhere (*Gurven et al., 2007*), including using known ages from written records, relative age lists, dated events, photo comparisons of people with known ages, and cross-validation of information from independent interviews of kin. Each method provides an independent estimate of age, and when estimates yielded a date of birth within a 3-year range, the average was generally used.

## Results
### Hunting cessation is associated with thoracic vertebral body fracture and, albeit weakly, with lower BMD

**Appendix 1—table 2.** Percentage of men (95% CI) who completely ceased hunting by age and thoracic vertebral body fracture status (fracture grade $\geq$2).

| Age category (years) | % ceased hunting with fracture | N | % ceased hunting without fracture | N |
|---|---|---|---|---|
| 40–49 | 0 (———) | 4 | 3 (<1–6) | 77 |
| 50–59 | 21 (<1–46) | 14 | 14 (6–22) | 78 |
| 60–69 | 60 (<1–1) | 5 | 32 (19–45) | 50 |
| 70+ | 100 (———) | 6[†] | 73 (53–93) | 22[‡] |
| Total | 41[**] (22–60) | 29 | 20 (15–25) | 227 |

^p$\leq$0.1, [*]p$\leq$0.05, [**]p$\leq$0.01, [***]p$\leq$0.001 ($\chi^2$ or Fisher's Exact Test [vs. no fracture]).
[†]Max age = 83 years (mean $\pm$ SD = 74 $\pm$ 4).
[‡]Max age = 83 years (mean $\pm$ SD = 74 $\pm$ 3).

**Appendix 1—table 3.** Binary logistic regression: effect of thoracic vertebral body fracture on the probability of hunting cessation after adjusting for age (model 1; n = 256 men).
Model 2 additionally includes as a covariate thoracic vertebral body bone mineral density (BMD). Shown are odds ratios (95% CIs); continuous variables are z-scored.

| Parameter | Model 1 | Model 2 (+BMD) |
|---|---|---|
| Any thoracic vertebral body fracture (grade ≥1; vs. none) | 7.3***(3.3–17.6) | 7.4***(3.3–18.2) |
| Age (years) | 5.2***(3.4–8.5) | 4.1***(2.5–7.0) |
| Thoracic vertebral body BMD (mg/cm$^3$) | ——— | 0.62*(0.38–0.99) |
| AIC | 178.55 | 176.61 |

^p≤0.1, *p≤0.05, **p≤0.01, ***p≤0.001 (refer to main text for FDR q-values)

## Hunting cessation associated with thoracic vertebral body fracture and lower BMD generates substantial productivity losses
Fracture

**Appendix 1—table 4.** Tsimane men's age-specific daily hunting production by thoracic vertebral body fracture status.

| Age (years) | (A) Hunt cals/day[†] | (B) Probability still hunting: no fracture[‡] | (C) Probability still hunting: fracture[‡] | (D) Hunt cals/day: no fracture[§] | (E) Hunt cals/day: fracture[¶] |
|---|---|---|---|---|---|
| 40 | 1872 | 0.995 | 0.968 | 1864 | 1812 |
| 41 | 1884 | 0.995 | 0.962 | 1874 | 1813 |
| 42 | 1888 | 0.994 | 0.955 | 1875 | 1803 |
| 43 | 1882 | 0.992 | 0.947 | 1868 | 1783 |
| 44 | 1870 | 0.991 | 0.938 | 1853 | 1754 |
| 45 | 1855 | 0.989 | 0.927 | 1835 | 1720 |
| 46 | 1836 | 0.987 | 0.915 | 1813 | 1680 |
| 47 | 1815 | 0.985 | 0.900 | 1787 | 1634 |
| 48 | 1790 | 0.982 | 0.884 | 1758 | 1582 |
| 49 | 1762 | 0.979 | 0.865 | 1725 | 1524 |
| 50 | 1731 | 0.975 | 0.843 | 1687 | 1459 |
| 51 | 1697 | 0.970 | 0.819 | 1646 | 1389 |
| 52 | 1659 | 0.965 | 0.792 | 1601 | 1314 |
| 53 | 1619 | 0.958 | 0.762 | 1551 | 1233 |
| 54 | 1575 | 0.951 | 0.729 | 1498 | 1148 |
| 55 | 1528 | 0.942 | 0.694 | 1440 | 1060 |
| 56 | 1478 | 0.932 | 0.656 | 1378 | 969 |
| 57 | 1425 | 0.920 | 0.616 | 1311 | 878 |
| 58 | 1369 | 0.906 | 0.574 | 1241 | 786 |
| 59 | 1310 | 0.891 | 0.531 | 1166 | 696 |
| 60 | 1247 | 0.873 | 0.488 | 1088 | 609 |
| 61 | 1181 | 0.852 | 0.445 | 1007 | 526 |
| 62 | 1113 | 0.829 | 0.403 | 923 | 449 |

*Continued on next page*

*Appendix 1—table 4 continued*

| Age (years) | (A) Hunt cals/day[†] | (B) Probability still hunting: no fracture[‡] | (C) Probability still hunting: fracture[‡] | (D) Hunt cals/day: no fracture[§] | (E) Hunt cals/day: fracture[¶] |
|---|---|---|---|---|---|
| 63 | 1041 | 0.803 | 0.362 | 836 | 377 |
| 64 | 966 | 0.775 | 0.323 | 748 | 313 |
| 65 | 888 | 0.743 | 0.287 | 660 | 255 |
| 66 | 807 | 0.709 | 0.253 | 572 | 204 |
| 67 | 723 | 0.672 | 0.222 | 486 | 160 |
| 68 | 636 | 0.633 | 0.193 | 402 | 123 |
| 69 | 546 | 0.592 | 0.168 | 323 | 92 |
| 70 | 452 | 0.549 | 0.145 | 249 | 66 |
| 71 | 356 | 0.506 | 0.125 | 180 | 44 |
| 72 | 257 | 0.463 | 0.107 | 119 | 28 |
| 73 | 154 | 0.421 | 0.092 | 65 | 14 |
| 74 | 49 | 0.379 | 0.078 | 19 | 4 |
| 75 | 0 | 0.339 | 0.067 | 0 | 0 |
| 76 | 0 | 0.302 | 0.057 | 0 | 0 |
| 77 | 0 | 0.267 | 0.048 | 0 | 0 |
| 78 | 0 | 0.234 | 0.041 | 0 | 0 |
| 79 | 0 | 0.205 | 0.035 | 0 | 0 |
| 80 | 0 | 0.178 | 0.029 | 0 | 0 |

[†]Predicted values (loess fit). From Jan 2005 to Dec 2009 adults were interviewed once or twice per week about time allocation and production for each co-resident individual over age 6 during the previous 2 days (n = 1245 individuals from 11 villages).

[‡]Predicted from binary logistic regression of whether one still hunts on thoracic vertebral body fracture status (i.e. grade ≥1; vs. none) and age.

[§]Derived by multiplying value in column A by value in column B.

[¶]Derived by multiplying value in column A by value in column C.

## Lower BMD

**Appendix 1—table 5.** Tsimane men's age-specific daily hunting production by thoracic vertebral body fracture status and thoracic vertebral body bone mineral density (BMD).
For illustrative purposes we report estimates holding BMD at +1 SD and −1 SD of the mean.

| Age (years) | (A) Hunt cals/day[*] | (B) Probability still hunting: no fracture and +1 SD BMD[†] | (C) Probability still hunting: no fracture and −1 SD BMD[†] | (D) Probability still hunting: fracture and +1 SD BMD[†] | (E) Probability still hunting: fracture and −1 SD BMD[†] | (F) Hunt cals/day: no fracture and +1 SD BMD[‡] | (G) Hunt cals/day: no fracture and −1 SD BMD[§] | (H) Hunt cals/day: fracture and +1 SD BMD[¶] | (I) Hunt cals/day: fracture and −1 SD BMD[**] |
|---|---|---|---|---|---|---|---|---|---|
| 40 | 1872 | 0.996 | 0.990 | 0.971 | 0.930 | 1865 | 1853 | 1819 | 1741 |
| 41 | 1884 | 0.995 | 0.988 | 0.967 | 0.920 | 1875 | 1862 | 1822 | 1732 |
| 42 | 1888 | 0.995 | 0.987 | 0.962 | 0.908 | 1878 | 1863 | 1816 | 1714 |
| 43 | 1882 | 0.994 | 0.984 | 0.956 | 0.895 | 1870 | 1853 | 1800 | 1684 |
| 44 | 1870 | 0.993 | 0.982 | 0.950 | 0.880 | 1857 | 1836 | 1776 | 1646 |
| 45 | 1855 | 0.992 | 0.979 | 0.942 | 0.864 | 1840 | 1816 | 1747 | 1602 |

*Continued on next page*

*Appendix 1—table 5 continued*

| Age (years) | (A) Hunt cals/day[*] | (B) Probability still hunting: no fracture and +1 SD BMD[†] | (C) Probability still hunting: no fracture and −1 SD BMD[†] | (D) Probability still hunting: fracture and +1 SD BMD[†] | (E) Probability still hunting: fracture and −1 SD BMD[†] | (F) Hunt cals/day: no fracture and +1 SD BMD[‡] | (G) Hunt cals/day: no fracture and −1 SD BMD[§] | (H) Hunt cals/day: fracture and +1 SD BMD[¶] | (I) Hunt cals/day: fracture and −1 SD BMD[**] |
|---|---|---|---|---|---|---|---|---|---|
| 46 | 1836 | 0.990 | 0.976 | 0.933 | 0.845 | 1819 | 1792 | 1714 | 1552 |
| 47 | 1815 | 0.989 | 0.972 | 0.924 | 0.825 | 1795 | 1764 | 1676 | 1497 |
| 48 | 1790 | 0.987 | 0.968 | 0.913 | 0.802 | 1767 | 1732 | 1633 | 1436 |
| 49 | 1762 | 0.985 | 0.963 | 0.900 | 0.778 | 1736 | 1697 | 1586 | 1371 |
| 50 | 1731 | 0.983 | 0.957 | 0.886 | 0.751 | 1701 | 1657 | 1533 | 1301 |
| 51 | 1697 | 0.980 | 0.951 | 0.870 | 0.723 | 1664 | 1614 | 1476 | 1226 |
| 52 | 1659 | 0.977 | 0.943 | 0.852 | 0.692 | 1621 | 1565 | 1414 | 1148 |
| 53 | 1619 | 0.974 | 0.935 | 0.833 | 0.660 | 1576 | 1514 | 1348 | 1068 |
| 54 | 1575 | 0.970 | 0.925 | 0.811 | 0.626 | 1527 | 1458 | 1278 | 986 |
| 55 | 1528 | 0.965 | 0.915 | 0.788 | 0.591 | 1474 | 1397 | 1203 | 902 |
| 56 | 1478 | 0.960 | 0.902 | 0.762 | 0.554 | 1418 | 1334 | 1126 | 819 |
| 57 | 1425 | 0.953 | 0.888 | 0.734 | 0.518 | 1359 | 1266 | 1046 | 738 |
| 58 | 1369 | 0.946 | 0.873 | 0.704 | 0.481 | 1296 | 1195 | 964 | 658 |
| 59 | 1310 | 0.938 | 0.856 | 0.672 | 0.444 | 1229 | 1121 | 881 | 582 |
| 60 | 1247 | 0.929 | 0.836 | 0.639 | 0.408 | 1159 | 1043 | 797 | 509 |
| 61 | 1181 | 0.919 | 0.815 | 0.604 | 0.373 | 1085 | 963 | 714 | 440 |
| 62 | 1113 | 0.907 | 0.792 | 0.568 | 0.339 | 1010 | 881 | 633 | 377 |
| 63 | 1041 | 0.894 | 0.766 | 0.532 | 0.307 | 931 | 798 | 554 | 319 |
| 64 | 966 | 0.879 | 0.739 | 0.495 | 0.276 | 849 | 714 | 478 | 267 |
| 65 | 888 | 0.862 | 0.709 | 0.458 | 0.248 | 766 | 630 | 407 | 220 |
| 66 | 807 | 0.844 | 0.678 | 0.422 | 0.221 | 681 | 547 | 340 | 178 |
| 67 | 723 | 0.823 | 0.645 | 0.386 | 0.197 | 595 | 466 | 279 | 142 |
| 68 | 636 | 0.801 | 0.610 | 0.352 | 0.174 | 509 | 388 | 224 | 111 |
| 69 | 546 | 0.776 | 0.575 | 0.319 | 0.154 | 424 | 314 | 174 | 84 |
| 70 | 452 | 0.750 | 0.538 | 0.288 | 0.136 | 339 | 243 | 130 | 61 |
| 71 | 356 | 0.721 | 0.501 | 0.258 | 0.119 | 257 | 178 | 92 | 42 |
| 72 | 257 | 0.690 | 0.464 | 0.231 | 0.105 | 177 | 119 | 59 | 27 |
| 73 | 154 | 0.658 | 0.428 | 0.206 | 0.092 | 101 | 66 | 32 | 14 |
| 74 | 49 | 0.624 | 0.392 | 0.183 | 0.080 | 31 | 19 | 9 | 4 |
| 75 | 0 | 0.588 | 0.358 | 0.162 | 0.070 | 0 | 0 | 0 | 0 |
| 76 | 0 | 0.552 | 0.324 | 0.143 | 0.061 | 0 | 0 | 0 | 0 |
| 77 | 0 | 0.515 | 0.293 | 0.125 | 0.053 | 0 | 0 | 0 | 0 |
| 78 | 0 | 0.479 | 0.263 | 0.110 | 0.046 | 0 | 0 | 0 | 0 |
| 79 | 0 | 0.442 | 0.236 | 0.096 | 0.040 | 0 | 0 | 0 | 0 |
| 80 | 0 | 0.406 | 0.210 | 0.084 | 0.035 | 0 | 0 | 0 | 0 |

[*]Predicted values (loess fit). From Jan 2005 to Dec 2009 adults were interviewed once or twice per week about time allocation and production for each co-resident individual over age 6 during the previous 2 days (n = 1245 individuals from 11 villages).

[†]Predicted from binary logistic regression of whether one still hunts on thoracic vertebral body fracture status (i.e. grade ≥1; vs. none), BMD (mg/cm$^3$) and age (years).

‡Derived by multiplying value in column A by value in column B.

§Derived by multiplying value in column A by value in column C.

¶Derived by multiplying value in column A by value in column D.

**Derived by multiplying value in column A by value in column E.

## Tree chopping cessation is associated with thoracic vertebral body fracture but not lower BMD

**Appendix 1—table 6.** Binary logistic regression: effect of thoracic vertebral body fracture on the probability of tree chopping cessation after adjusting for age (model 1; n = 256 men).
Model 2 additionally includes as a covariate thoracic vertebral body bone mineral density (BMD). Shown are odds ratios (95% CIs); continuous variables are z-scored.

| Parameter | Model 1 | Model 2 (+BMD) |
|---|---|---|
| Any thoracic vertebral body fracture (grade ≥1; vs. none) | 6.9***(3.1–16.6) | 6.8***(3.1–16.6) |
| Age (years) | 8.0***(4.9–13.9) | 6.8***(4.0–12.4) |
| Thoracic vertebral body BMD (mg/cm³) | ——— | 0.75 (0.47–1.16) |
| AIC | 183.41 | 183.75 |

ˆp≤0.1, *p≤0.05, **p≤0.01, ***p≤0.001 (refer to main text for FDR q-values).

## Weaving cessation is not associated with thoracic vertebral body fracture but is associated, albeit weakly, with lower BMD

**Appendix 1—table 7.** Binary logistic regression: effect of thoracic vertebral body fracture on the probability of weaving cessation after adjusting for age (model 1; n = 237 women).
Model 2 additionally includes as a covariate thoracic vertebral body bone mineral density (BMD). Shown are odds ratios (95% CIs); continuous variables are z-scored.

| Parameter | Model 1 | Model 2 (+BMD) |
|---|---|---|
| Any thoracic vertebral body fracture (grade ≥1; vs. none) | 2.2 (0.8–6.4) | 1.8 (0.6–5.4) |
| Age (years) | 4.9***(3.1–8.6) | 3.3***(1.8–6.6) |
| Thoracic vertebral body BMD (mg/cm³) | ——— | 0.51ˆ(0.23–1.05) |
| AIC | 134.52 | 133.18 |

ˆp≤0.1, *p≤0.05, **p≤0.01, ***p≤0.001 (refer to main text for FDR q-values).

**Appendix 1—table 8.** Self-reported reasons for weaving cessation by thoracic vertebral body fracture status.
Shown are percentages of women reporting a given reason.

| Reason for weaving cessation | Fracture (n = 9) | No fracture (n = 24) | Total (n = 33) |
|---|---|---|---|
| Problem with hips | 100 | 100 | 100 |
| Problem with back | 100 | 100 | 100 |
| Problem with fingers | 100 | 100 | 100 |

*Continued on next page*

*Appendix 1—table 8 continued*

| Reason for weaving cessation | Fracture (n = 9) | No fracture (n = 24) | Total (n = 33) |
|---|---|---|---|
| Problem with hands (other than fingers) | 100 | 100 | 100 |
| Any problem with hips, back, or hands | 100 | 100 | 100 |
| Difficulty sitting | 100 | 100 | 100 |
| Difficulty seeing | 89 | 88 | 88 |

^p≤0.1, *p≤0.05, **p≤0.01, ***p≤0.001 ($\chi^2$ or Fisher's Exact test [vs. no fracture]).

## Restricted day range is associated with thoracic vertebral body fracture and, albeit weakly, with lower BMD

**Appendix 1—table 9.** Binary logistic regression: effect of thoracic vertebral body fracture on the probability of not being able to walk a full day after adjusting for age and sex (model 1; n = 493 adults).
Model 2 additionally includes as a covariate thoracic vertebral body bone mineral density (BMD). Shown are odds ratios (95% CIs); continuous variables are z-scored. Interaction terms between sex and either fracture, age or BMD do not yield significant parameter estimates and are not shown.

| Parameter | Model 1 | Model 2 (+BMD) |
|---|---|---|
| Any thoracic vertebral body fracture (grade ≥1; vs. none) | 8.2***(4.8–14.5) | 7.8***(4.5–13.8) |
| Age (years) | 4.0***(3.1–5.4) | 3.4***(2.4–4.8) |
| Sex = male | 0.19***(0.11–0.31) | 0.21***(0.12–0.34) |
| Thoracic vertebral body BMD (mg/cm$^3$) | ——— | 0.77(0.57–1.03) |
| AIC | 479.60 | 478.52 |

^p≤0.1, *p≤0.05, **p≤0.01, ***p≤0.001 (refer to main text for FDR q-values).

## Suggestive evidence of fracture preceding task cessation, rather than the reverse

**Appendix 1—table 10.** Self-reported reasons for hunting cessation by thoracic vertebral body fracture status.
Shown are percentages of men reporting a given reason.

| Reason for hunting cessation | Fracture (n = 34) | No fracture (n = 22) | Total (n = 56[†]) |
|---|---|---|---|
| Problem with hips | 29**‡ | 0 | 18 |
| Problem with back | 53 | 59 | 55 |
| Problem with arms | 12 | 23 | 16 |
| Problem with legs | 56 | 55 | 55 |
| Any problem with hips, back, or limbs | 77 | 86 | 80 |
| Feels weak | 29^ | 50 | 38 |
| Tires easily | 59 | 64 | 61 |
| Tires easily or feels weak | 65 | 77 | 70 |
| Difficulty hearing | 38 | 50 | 43 |
| Difficulty seeing | 62 | 77 | 68 |
| Difficulty hearing or seeing | 65 | 82 | 71 |

ˆp≤0.1, *p≤0.05, **p≤0.01, ***p≤0.001 ($\chi^2$ or Fisher's Exact test [vs. no fracture]).

[†]Fracture data are missing for one man. For another man self-reported data are missing.

[‡]FDR q = 0.012.

Note: only q-values ≤0.05 are reported.

**Appendix 1—table 11.** Self-reported reasons for tree chopping cessation by thoracic vertebral body fracture status.

Shown are percentages of men reporting a given reason.

| Reason for tree chopping cessation | Fracture (n = 41) | No fracture (n = 35) | Total (n = 76[†]) |
|---|---|---|---|
| Problem with hips | 29*[‡] | 9 | 20 |
| Problem with back | 59 | 71 | 64 |
| Problem with arms | 66ˆ | 83 | 74 |
| Problem with legs | 29 | 26 | 28 |
| Any problem with hips, back, or limbs | 83 | 94 | 88 |
| Feels weak | 54 | 54 | 54 |
| Tires easily | 63 | 71 | 67 |
| Tires easily or feels weak | 80 | 83 | 82 |

ˆp≤0.1, *p≤0.05, **p≤0.01, ***p≤0.001 ($\chi^2$ or Fisher's Exact test [vs. no fracture]).

[†]Fracture data are missing for one man. For another man self-reported data are missing.

[‡]FDR q = 0.036.

Note: only q-values ≤0.05 are reported.

**Appendix 1—table 12.** Self-reported reasons for inability to walk all day by thoracic vertebral body fracture status.

Shown are percentages of adults (pooled sexes) reporting a given reason.

| Reason for inability to walk all day | Fracture (n = 95) | No fracture (n = 134) | Total (n = 229[†]) |
|---|---|---|---|
| Problem with hips | 39 | 39 | 39 |
| Problem with back | 61 | 69 | 66 |
| Problem with arms | 8 | 8 | 8 |
| Problem with legs | 81 | 86 | 84 |
| Any problem with hips, back, or limbs | 88 | 93 | 91 |
| Feels weak | 29 | 31 | 31 |
| Tires easily | 71 | 60 | 65 |
| Tires easily or feels weak | 76 | 69 | 72 |

ˆp≤0.1, *p≤0.05, **p≤0.01, ***p≤0.001 ($\chi^2$ or Fisher's Exact test [vs. no fracture]).

[†]Fracture data are missing for one adult.

