## [Decision Letter]

**Acceptance summary:**

This study examines relationships among bone fragility, disability, and subsistence costs in a forager-horticulturalist society, the Tsimane (Bolivia). Vertebral fracture and lower bone mineral density are associated with substantial productivity losses from the cessation of hunting, tree chopping, and long-distance walking activities. These results help improve our understanding of the potential complex selective pressures associated with human longevity and our species' propensity for age-related fragility.

**Decision letter after peer review:**

[Editors’ note: the authors submitted for reconsideration following the decision after peer review. What follows is the decision letter after the first round of review.]

Thank you for submitting your work entitled "Productivity loss associated with physical impairment in a contemporary small-scale subsistence population" for consideration by *eLife*. Your article has been reviewed by three peer reviewers, including George H Perry as the Senior and Reviewing Editor and Reviewer #1, The following individual involved in review of your submission has agreed to reveal their identity: Susan Pfeiffer (Reviewer #3).

Our decision has been reached after consultation between the reviewers. Based on these discussions and the individual reviews below, we regret to inform you that your work will not be considered further for publication in *eLife* at this time. I do not want to completely close the door on our future consideration of this work, as we truly liked much about it and we believe that it has much promise. However, the consensus outcome of our reviewer consultation process was that at the least some major core work is necessary that likely exceeds the scope of revision acceptable in the *eLife* model, and resulting in substantial changes to the paper requiring a new full review if you were to choose to submit a new manuscript to *eLife*. Note that while the reviewers are enthusiastic in some respects they wanted to emphasize the fundamental nature of some of the concerns that they feel need to be fully considered and addressed.

Reviewer #1:

I'm reviewing this manuscript from a general biological anthropology perspective only, without direct expertise in bone health measures and interpretations of those data, so I will default to other reviewers accordingly.

I thought that the overall study was fairly solid and insightful. The authors integrate ethnographic data on Tsimane activity and economy with thoracic vertebrae bone mineral density estimates (and fracture incidence) from computed tomagraphy (CT) scans for a large sample of n=493 individuals.

1) The beginning investigations of correlation vs. causation are valuable and interesting, but I do not think that they are definitive to warrant the causative statements made by the authors throughout the paper. I think that such statements should be changed to “associated with”, with discussion of correlation/causation supported by the post-hoc analyses that help in this direction (but are not definitive, given so many other things that cannot be accounted for) in the Discussion.

2) I don't agree with the practice applied throughout the paper of discretizing the data by age categories and positive versus negative BMD residuals, etc., for analyses and visualization. Such data-reducing discretization should be limited as much as possible; please analyze and show the full data. Also, anytime histograms are necessary or useful for visualization, use jitter to show the underlying data points.

3) The multiple testing burden is also not being considered at present. The authors should compute false discovery rate q-values for all of the tests conducted within each major hypothesis test and report these as the main probability estimate.

4) I would also like to have a visualization of the hunting activity data alongside the BMD and fracture data as a function of age. Presenting the results of the models are fine but these are the critical data of the study and there needs to be an accompanying strong data visualization. What is being shown in Figures 3, 4, and 5? If these are summary models they should only be shown as supporting panels to main figures with the actual data. The quality of the figures in general in the paper needs to be improved.

Reviewer #2:

The evolutionary hypotheses are of interest with relevant objectives. The statistical analyses are rigorous and correctly utilized.

However, there are several key issues with the overarching design and interpretation of the study methods (bone health as a proxy for impairment) and wider contextualization of archaeological and evolutionary studies of bone health that are needed in this study in order to be published.

Biological contextualization and understanding of the method of bone analysis is lacking, and discussion of the finding of poor bone health (low bmd, fracture in youngish age males) needs to be given. While I realize this (bone biology/osteoporosis data and interpretation) is not the focus of this study, not including this discussion/data causes the study design and method to appear flawed, and in many places the paper has sweeping statements that are simply incorrect. The measure of BMD is a crude measure, and data for different groups (sexes, and ages) are likely needed to be considered or discussed separately as they have likely different etiologies and different outcomes/costs because of this. Readership is perhaps better suited to a specialty journal (ex Bone, JBMR) but this data in current form is not rigorous to be published there.

Specifically:

Adult physical impairment (API) is described as degenerative or other origin (e.g. trauma, infection). But this is a very wide statement – this includes a multitude of pathological conditions, that are obviously part of the human (long-lived vertebrate/mammalian?) condition – so of course these things will hinder foraging in any animal. The authors go on to cite “specifically” degenerative conditions that are osteoarthritic as things that are API in human evolution, and specifically cause back pain (the closest link to this study) but then many other conditions and diseases (trauma, infectious, degenerative, developmental….):

1) There is an entire discipline (disability studies) and a multitude of social science data studies that have critiqued the use of all disease and all pathology as indicative of impairment, or worse the use any disease, as a proxy for lack of productivity and/or ability. I strongly suggest the authors "pick" a disease to build a better case for why this maker (in this case low bmd and spinal fracture) can be used as indicative of impairment.

2) The authors do not build legitimate case or correct lit review that low bmd and mild fracture is related to back pain. In fact, until there is severe vertebral collapse the correlation to clinical back pain is not usually made and is not consistent. What is linked to back pain and impairment very clearly is spinal osteoarthritis – a pathology that has nothing to do with BMD or osteopenic fracture. The authors go on to cite in fact spinal arthritis studies that found in hominin studies (Chapman; Cook; Trinkaus etc);

More inconsistencies in data design and discussion:

Why are thoracic vertebrae "are more common among Tsimane than matched US (Los Angeles) controls "? It is not true that thoracic fractures are the most common in human populations – lower thoracic and upper lumbar are. Is this why the Tsimane look higher %- as in US controls lumbar fractures are higher? Why did the authors not look at lumbar vertebra: The CT work in this study was part of an atherosclerosis study of the thoracic cage- so lumbar vertebrae that are the most diagnostic of bone loss impairment we not included in this study.

It is relevant to know why Tsimane males in particular have low bmd and thoracic fracture – this is not an expected age-related phenomena. Is the low bmd a sign of another causative pathology that is causing the impairment of activity and pain? If so, low bmd is only a secondary measure of an underlying impairment that may not have to do with bone health at all – maybe dietary, parasitic, related to blood disorder that causes bone loss and impairment? Is this not relevant to discuss or mention?

What is going on with male bone loss and female bone loss is clearly different, as discussed by authors that point to previous studies that show female bone loss is related to post-menopausal or reproductive reasons. And both sexes show different "costs" of the bone health measure as API – they sexes needed to be discussed differently – and again why the "cause” of the male bone loss is important to explain. They likely have different etiology in both sexes, different costs associated because of this.

Again wide sweeping statements with no basis for any meaningful interpretation: "Sixty-eight percent of adults age 50+ are also clinically diagnosed with some musculoskeletal problem by Tsimane Health and Life History Project (THLHP)" and "some physical discomfort during activities". – what is "musculoskeletal problem" and how can we use this as an equivalent of the BMD measure used in this study?

Statements that are not true or well researched/cite:

"Ample bioarchaeological evidence of partially or fully healed fractures (e.g. (Lovejoy and Heiple, 1981; Pfeiffer, 2012)) and developmental abnormalities (Cowgill et al., 2015; Trinkaus, 2018) suggests longer-term survival with varying levels of disability.” In fact, numerous bioarcheological studies have shown that “osteopenic” fractures with low BMD are very rare in the past – traumatic/violent fractures are very common

"Tsimane thoracic vertebral fractures appear to result from both trauma and senescence (Stieglitz et al., 2019) and are not just prevalent among the oldest adults.”

So are we looking at violence or accidental trauma fracture in males here – or related to another primary disease? This needs to be discussed/expanded. It is not senescence in young age males.

Again – both sexes -age groups have different etiologies of low BMD and likely different outcomes/costs because of this.

Reviewer #3:

The submitted manuscript describing productivity loss linked to bone mass decline is well written, novel and pertinent in several domains. It is certainly a worthwhile contribution. The suggestions made here for adjustments and improvements reflect this reviewer's responses to the work. The clear statements at beginning and end regarding the importance of this kind of research for bioarchaeological and paleoanthropological studies is welcome.

Perhaps the broadest concern with the current version of the manuscript is the tendency to under-play the imbalance of impact of bone mass loss between men and women. Clearly, the paper's focus is on hunting and fishing, which is all good. However, from Abstract to conclusions the different pattern in women is underplayed. The type of loom(s) used by the weavers could be pertinent to the study and should be specified. The brief section on the lack of significant effect on weaving should be moved up so that it precedes the section on long distance walking. The relative inattention to women's productivity is a related concern. There should be acknowledgment, as the authors discuss the limitations of the work, of the fact that their data are at least somewhat inconsistent with modern industrialized countries' clinical data, with regard to sex patterns in BMD and the disability load associated with compression fractures.

The research presented here links to a research area of which the authors appear to be unaware. In a 1996 paper (Anatomical Record 246:4:423-432), Agarwal and Grynpas asserted that osteoporotic fracture is absent in past populations. While it has only been cited 41 times, those citations include recent work. I urge the authors to become better informed on just how novel – and helpful – their work is, in a bioarchaeological context. In my opinion, the best explanation for the "pattern" reported in that 1996 paper is that no one ever systematically studies thoracic compression fractures. Nevertheless, there is a literature arguing that osteoporosis is a modern disease (cf. Kralick and Zemel, 2020 Frontiers in Endocrinology).

With regard to which vertebrae form the basis of the dataset: If T7 or T8 form the starting point for the assessment, the technician might be providing bone mass input from groups as different as T5-T7 to T8-T10. These are very different parts of the thoracic column, from the bone mass perspective. This is a concern that should be addressed, if possible. The morphological assessment of T6 to T12 helps to give the reader more confidence in the data.

The discussion of the contributions of older group members would benefit from stretching beyond physical activities to intellectual assets, along the lines of D.J. Levitin's argument that older adults contribute experience and judgement. The impressive breadth of the framework constructed by the authors could and should be even broader and more interdisciplinary.

[Editors’ note: further revisions were suggested prior to acceptance, as described below.]

Thank you for submitting your article "Productivity loss associated with physical impairment in a contemporary small-scale subsistence population" for consideration by *eLife*. Your article has been reviewed by two peer reviewers, and the evaluation has been overseen by George Perry as the Senior and Reviewing Editor. The following individual involved in review of your submission has agreed to reveal their identity: Ian Wallace (Reviewer #1).

The reviewers have discussed the reviews with one another and the Senior and Reviewing Editor has drafted this decision to help you prepare a revised submission.

Due to pandemic-related constraints, the two reviewers besides myself of your related manuscript submission were not able to reconsider this new version. However, each of the two new reviewers were made aware of the previous round of review, had access to your response to the previous reviewer comments, and I took this factor into consideration for this decision letter. I also did re-review the paper myself, though I did not write a formal review given that you had addressed my most substantive comments in your new submission.

One major concern about your previous submission had been the need to avoid conflating association and causation. I noticed how you quite thoroughly addressed this concern in the Results section of your revised manuscript. Thank you. However, interestingly, this was still the overarching major concern noted by the two new reviewers, with this concern now focused on the Discussion section and your interpretations. We had a great consultation discussion on this issue. We collectively do find much to like about the study and paper, but agree that this overarching concern needs to be resolved robustly. The following is a summary of the essential revisions, with the detailed reviewer comments appended below for expanded direction and explicit instruction:

The authors need to 1) reduce statements of implied causation in the discussion (and also in the general setup of the paper regarding the possible causal links between bone fragility and disability) and fully acknowledge all possibilities and 2) carefully establish potential mechanisms for any proposed causality, even when reduced explicitly to speculation. Currently the discussion is written as if the correlation is directional and infers causation, but with so many unstudied covariates (since lower BMD is a function of aging) this is inappropriate. The appropriate null hypothesis for interpretation should be carefully considered. I support some hypothesis generation and speculation in the discussion but this should be clearly indicated as such, and clearly separated from the conclusive interpretations. Even as such, the potential mechanistic links here would need to be much more carefully detailed and supported given the reliance on indirect inference only and the inconsistency with existing literature. Please see the reviewer comments for explicit suggestions and details.

Reviewer #1:

This is a very nice study of associations between bone fragility, disability, and subsistence costs among a small-scale forager-horticulturalist society, the Tsimane of Bolivia. The study builds on a great recent paper in *eLife* by the same authors showing that the Tsimane (especially women) have lower average vertebral BMD values than might be expected given their high physical activity levels, as well as surprisingly high rates of vertebral fractures. In this study, they investigate the impact of low vertebral BMD and fracture incidence on disability (or, "adult physical impairment" as they call it) and ultimately the economic costs of such disability. The paper nicely frames the study within evolutionary theory and provides novel insights on the selective pressures that may have shaped human longevity and our species' propensity for age-related fragility. The sample analyzed is impressively large, particularly for a remote-living population like the Tsimane. The CT methods used to collect the bone data are the gold standard. The paper is clear and well written, and the interpretations of the results are all reasonable. Also, the figures are nice. Overall, I think this paper deserves consideration for publication in *eLife*, but I have a couple of concerns.

First, in my opinion, the possible causal links between bone fragility and disability are not considered or described in enough detail. The only mention of such a link is: "Vertebral deformities are reliably associated with chronic pain…(no citation)" Do the authors have data on back pain? If so, the link between bone fragility and disability would be a lot stronger if they could show that bone fragility is indeed associated with chronic pain among the Tsimane. If not, then it remains possible that bone fragility and disability are only indirectly related. For example, perhaps in many cases bone fragility is just one manifestation of a more general somatic decline, and disability is engendered primarily by other aspects of the general phenomenon and not bone fragility per se. If the authors do not have pain data, I suggest they at least make a stronger argument for the link between bone fragility and pain/disability by citing prior research demonstrating such links.

Second, on the same topic of causal links between bone fragility and disability, I believe the authors must provide a more detailed explanation for how they interpret the contribution of low BMD to disability. Independent of a fracture, there is no clear reason why one would expect low BMD to directly affect disability. Again, it is possible that low BMD and disability could be indirectly related if low BMD is only one of multiple outcomes of general somatic decline. But my understanding is that osteoporosis is generally an asymptomatic disease until fracture occurs. My impression is that the authors may not necessarily be arguing that low BMD, independent of fracture, affects disability, but rather that fracture plus low BMD leads to more disability than a fracture alone. Even so, why does this scenario make sense? Are fractures experienced by a bone with low BMD inherently more painful? Maybe the cooccurrence of fracture and low BMD leads to more disability because such fractures are generally more severe. Or, given that bone fractures can trigger bone loss, maybe the cooccurrence of fractures and low BMD represent cases where fractures took place longer ago and thus potentially have a greater effect on chronic pain and disability. An extreme way of resolving this uncertainty would be for the authors to disregard the BMD data entirely and focus just on the fracture data. The overall storyline would not change too much. However, I am not suggesting the authors do this. The BMD data are potentially important. But the links with disability need to be more carefully considered.

Reviewer #2:

This paper aims to estimate the magnitude of productivity losses associated with compromised bone strength among the Tsimane. The authors specifically test "whether a specific morphological characteristic which may indicate physical impairment, i.e. compromised bone strength, is indeed associated with diminished participation in routine subsistence tasks". They "identify economic correlates of compromised bone strength (i.e. thoracic vertebral body fracture and lower thoracic vertebral BMD) by examining involvement in four common tasks in foraging economies: hunting, tree chopping, tool manufacture (i.e. weaving), and walking long distances".

The data presented in this paper are important and have the potential to make a valuable contribution to the literature, and the authors are to be congratulated for conducing important, unique, and valuable fieldwork. The questions they have posed in this paper are interesting and worthy of consideration. That being said, I have some concerns with the analysis, what it assumes, what it excludes, and the conclusions.

The paper is set to test the "probability of completely ceasing to perform a given task increases with compromised bone strength", and estimate the "magnitude of productivity losses (i.e. lost kcals/day) from hunting cessation associated with compromised bone strength" The authors claim "We do not assume that compromised bone strength is itself a proxy for functional impairment or reduced productivity. Instead we posit that individuals with compromised bone strength who are no longer able to perform specific subsistence tasks facultatively adjust their time allocation based on various factors (e.g. phenotypic condition, household demand) to maximize productivity." This is really unclear and woolly. What, specifically then, are they predicting? If they don't think compromised bone strength influences productivity, why are they testing it, and why is the text replete with indications that they are interpreting this as a direct influence?

The authors make their assumptions of the direction of this relationship clear throughout the paper, through the use of language such as "To estimate men's hunting productivity losses associated with fracture and lower BMD". The paper is replete with the inference of this relationship, and while the authors might carefully choose their words in places to try to avoid causation, the inference is there for the reader to make. Perhaps another way of putting this is the following: the authors show correlations between measures of later life activity/productivity and both increase fracture prevalence and lower BMD. There are three parsimonious explanations for this correlation:

1) The loss of bone quality is the cause of a decline in involvement in economic activity

2) The loss of bone quality is correlated with age, which independently corresponds with later life declines in economic activity, and the two are not directly related but collinear

3) A later life decline in activity is the cause of a decline in bone quality.

These are simplistic explanations, and there are likely other factors involved (see comments below), but for the purposes of simplicity, let's consider the above. The language of the paper is only oriented around the first explanation and the direction of causality is assumed through most of the text, but only really directly acknowledged when they briefly address the possibility of “reverse causality” (reductions in activity causing a reduction in bone strength). They attempt to test for “reverse causality” through indirect methods: comparing fracture status to self-reported reasons for task cessation, and the use of GLM to compare BMD of those who cease versus those who continue participation. I'm not convinced that these adequately address the challenges of the direction of the relationship. Self-reported causes may, in some cases, related to injury (particularly in severe cases), but are much more likely to relate to other factors entirely, and to be unrelated to BMD. The GLM comparisons of BMD among groups defined by continued participation in activity may be influenced by study design, and I am not certain it is testing the reverse causation that the authors suggest. The most significant influence on later life bone mass is earlier life activity. Peak bone mass is commonly achieved between the ages of 20 and 30, and is the single most important predictor of later life bone mass. The more you build early on, the more bone you have to lose. Activity at age 20 may thus be stronger correlate of bone mass after the age of 40 than recent activity patterns. It is curious that the authors point out that "Physical strength peaks in the mid to late 20s, then declines continuously thereafter, reaching ~60% of maximum adult capacity by age 70 (Gurven et al., 2006)." This is essentially the same trajectory of bone strength acquisition and loss, and there is considerable evidence that bone strength corresponds with muscle induced loads. One could argue the most parsimonious explanation is that the most active and economically productive individuals during young adulthood have the highest peak bone mass, and are more likely to remain productive (or with higher than average productivity) later in life.

Apart from the issues noted above, I remain sceptical about causation. There is, in theory, a direct causal relationship between fractures and decline or cessation of economic activity. This is supported by the relatively strong results, for example, the declines in weaving and ability to walk for a day associated with earlier fractures. What is the causal mechanism for the relationship with BMD? Going back to the three possible interpretations I mentioned earlier, I would have no issue with the statement of a hypothesis that vertebral fracture is the cause of a decline in involvement in economic activity. If I were to hypothesize what would explain the correspondence with BMD, I would go for a variant of the second hypothesis, that it is correlated with age, which independently correlates with other measures that are associated with later life declines. There are multiple collinear biological trends associated with senescence. While the authors provide no consideration of a biological mechanism through which BMD would be associated directly with productivity, the inference of causation is made without consideration of the particular relationship of declines in BMD associated with other factors of aging.

I am also sceptical of the consideration of both BMD and trauma as indicators of bone strength. BMD is highly correlated with bone mechanical function, this is uncontroversial, but considering trauma as a similar indicator may be problematic. Demographers and epidemiologists have often noted that trauma occurs in different societies with a relatively consistent frequency (at least when compared to infectious or metabolic disease, for example), because they result from accidental injury which tend to occur with some statistical regularity in larger populations. The data in Figure 2 suggest that there is no real difference the relationship between BMD and Age between groups of men with and without fracture, which could be interpreted as evidence that fracture prevalence among is predominantly determined by accidents rather than a loss of bone strength, regardless of the age or BMD of the individual. In contrast, there appear to be considerable differences among the women measured, suggesting that BMD and Fractures are more strongly correlated in women. The male data, at least, would seem to contradict the assertion that fracture prevalence is a proxy for bone strength.

---

## [Author Response]

[Editors’ note: the authors resubmitted a revised version of the paper for consideration. What follows is the authors’ response to the first round of review.]

Reviewer #1:I'm reviewing this manuscript from a general biological anthropology perspective only, without direct expertise in bone health measures and interpretations of those data, so I will default to other reviewers accordingly.I thought that the overall study was fairly solid and insightful. The authors integrate ethnographic data on Tsimane activity and economy with thoracic vertebrae bone mineral density estimates (and fracture incidence) from computed tomagraphy (CT) scans for a large sample of n=493 individuals.1) The beginning investigations of correlation vs. causation are valuable and interesting, but I do not think that they are definitive to warrant the causative statements made by the authors throughout the paper. I think that such statements should be changed to “associated with”, with discussion of correlation/causation supported by the post-hoc analyses that help in this direction (but are not definitive, given so many other things that cannot be accounted for) in the Discussion.

We have modified text accordingly (in the Introduction, Results and Discussion), so as not to imply that the study definitively establishes causality.

2) I don't agree with the practice applied throughout the paper of discretizing the data by age categories and positive versus negative BMD residuals, etc., for analyses and visualization. Such data-reducing discretization should be limited as much as possible; please analyze and show the full data. Also, anytime histograms are necessary or useful for visualization, use jitter to show the underlying data points.

We revised all figures based on both this comment and Dr. Perry’s comment below (comment #4). We no longer discretize data by age categories or BMD residuals, and we now show the full raw data for primary bone strength measures (i.e. fracture and BMD; see Figures 2-3).

3) The multiple testing burden is also not being considered at present. The authors should compute false discovery rate q-values for all of the tests conducted within each major hypothesis test and report these as the main probability estimate.

We agree that it is important to consider the multiple testing burden for certain analyses, and appreciate the recommendation. We now report FDR q-values in the revision (in the text and Appendix 1—tables 10-11).

4) I would also like to have a visualization of the hunting activity data alongside the BMD and fracture data as a function of age. Presenting the results of the models are fine but these are the critical data of the study and there needs to be an accompanying strong data visualization. What is being shown in Figures 3, 4, and 5? If these are summary models they should only be shown as supporting panels to main figures with the actual data. The quality of the figures in general in the paper needs to be improved.

The quality of all figures has been enhanced by redoing all the figures from scratch. We agree that it is more intuitive and useful for the reader to show actual production data, rather than productivity differentials. Figures 4-5 now show actual hunting/fishing productivity estimates (both age-specific and expected cumulative future) by fracture status and BMD, rather than the difference in hunting productivity by fracture status and BMD. We do not include raw fracture and BMD data (now shown in Figures 2-3) in the same figure as hunting/fishing productivity data (now shown in Figures 4-5) because we lack all of these data at the individual-level (as explained in Materials and methods). The values shown in Figures 4-5 (reported in Appendix 1—tables 4-5) thus represent synthetic, model derived estimates rather than raw values. Lastly, we deleted the final figure in the original manuscript (old Figure 5) summarizing subsistence involvement (i.e. 1-probability of task cessation) across all tasks by age, fracture status and BMD because it was not critical to our central thesis, and to eliminate data-reducing discretization of BMD based on positive and negative residuals.

Reviewer #2:The evolutionary hypotheses are of interest with relevant objectives. The statistical analyses are rigorous and correctly utilized.However, there are several key issues with the overarching design and interpretation of the study methods (bone health as a proxy for impairment) and wider contextualization of archaeological and evolutionary studies of bone health that are needed in this study in order to be published.

We do not consider “bone health” as a proxy for functional ability or impairment: whether and to what extent compromised bone health is associated with impaired productivity are empirical questions which we address in this manuscript (without assuming an answer ex ante). We have clarified our position in the Introduction and Discussion to avoid potential misinterpretations.

Biological contextualization and understanding of the method of bone analysis is lacking, and discussion of the finding of poor bone health (low bmd, fracture in youngish age males) needs to be given.

We make no assessment or judgment about whether specific BMD values or fracture reflects “poor bone health” (we do not impose BMD thresholds [e.g. based on DXA-derived T-scores] denoting osteopenia or osteoporosis). Instead we empirically test for associations between subsistence involvement and bone strength measures, including continuous BMD values, and we use those parameter estimates in our calculations of productivity losses. To derive those calculations we make rather conservative assumptions regarding the relationship between compromised bone strength and productivity (e.g. we do not assume that fracture or lower BMD values are associated with lower foraging efficiency).

While I realize this (bone biology/osteoporosis data and interpretation) is not the focus of this study, not including this discussion/data causes the study design and method to appear flawed, and in many places the paper has sweeping statements that are simply incorrect.

We do not believe any statements in the revised manuscript are incorrect.

The measure of BMD is a crude measure, and data for different groups (sexes, and ages) are likely needed to be considered or discussed separately as they have likely different etiologies and different outcomes/costs because of this.

First, BMD accounts for ~70% of the variance in bone strength and is widely used in research and clinical settings to indicate bone strength. Second, the sexes are indeed analyzed separately, as the sexual division of labor would require. Third, we follow the recommendation of the Senior and Reviewing Editor and treat age as a continuous not categorical variable.

Readership is perhaps better suited to a specialty journal (ex Bone, JBMR) but this data in current form is not rigorous to be published there.

We respectfully disagree, and we thank the Senior and Reviewing Editor for providing us with the opportunity to resubmit.

Specifically:Adult physical impairment (API) is described as degenerative or other origin (e.g. trauma, infection). But this is a very wide statement – this includes a multitude of pathological conditions, that are obviously part of the human (long-lived vertebrate/mammalian?) condition – so of course these things will hinder foraging in any animal. The authors go on (p 4) to cite “specifically” degenerative conditions that are osteoarthritic as things that are API in human evolution, and specifically cause back pain (the closest link to this study) but then many other conditions and diseases (trauma, infectious, degenerative, developmental….):

We agree that this original description was too wide-ranging, and we thank reviewer #2 for pointing this out. We have narrowed the focus of the Introduction accordingly, so as not to detract attention from the potential impairments that we consider in this paper, i.e., arising from compromised bone strength. When other health conditions (e.g. infection) are mentioned in the Introduction, it is very brief and meant to acknowledge prior studies of extant subsistence populations that have similarly examined associations between somatic condition and productivity.

1) There is an entire discipline (disability studies) and a multitude of social science data studies that have critiqued the use of all disease and all pathology as indicative of impairment, or worse the use any disease, as a proxy for lack of productivity and/or ability. I strongly suggest the authors "pick" a disease to build a better case for why this maker (in this case low bmd and spinal fracture) can be used as indicative of impairment.

We do not view all disease and all pathology as indicative of impairment, nor do we view the use of any disease as a proxy for lack of productivity and/or ability. We never intended to imply this. Whether and to what extent a given condition is associated with impairment or reduced productivity is an empirical question, and one that our study is designed to address. We have clarified our position in the Introduction in light of reviewer #2’s comment.

2) The authors do not build legitimate case or correct lit review that low bmd and mild fracture is related to back pain. In fact, until there is severe vertebral collapse the correlation to clinical back pain is not usually made and is not consistent. What is linked to back pain and impairment very clearly is spinal osteoarthritis – a pathology that has nothing to do with BMD or osteopenic fracture. The authors go on to cite in fact spinal arthritis studies that found in hominin studies (Chapman; Cook; Trinkaus etc);

Back pain is not the focus of our study, and not something we are trying to explain as a direct result of lower BMD or fracture (mild or other). Back pain can also be prevalent and severe in adults whose bone strength is not compromised. Nevertheless, we do analyze data on self-reported reasons for subsistence task cessation, including back problems (Appendix 1—tables 8; 10-12). These data suggest few differences in the experience of back problems (including pain) for adults with vs. without fracture, and indicate a fairly high overall prevalence of back problems. Back pain may be one of several mechanisms linking thoracic vertebral fracture and diminished subsistence involvement.

More inconsistencies in data design and discussion:Why are thoracic vertebrae "are more common among Tsimane than matched US (Los Angeles) controls "?

The population-level difference in thoracic vertebral body fracture prevalence is now clarified in Materials and methods, Study population and Discussion, although an expanded explanation is beyond the scope of the present study (see Stieglitz et al., 2019, for an expanded explanation).

It is not true that thoracic fractures are the most common in human populations – lower thoracic and upper lumbar are.

We never said “the most common…” but instead we said “among the most common”, which is not inaccurate.

Is this why the Tsimane look higher %- as in US controls lumbar fractures are higher?

In Stieglitz et al., 2019 (see Figure 3) we report that Tsimane (both sexes) show higher age-standardized thoracic vertebral body fracture prevalence than Los Angeles residents, using directly comparable measures of thoracic vertebral fracture generated by the same lab.

Why did the authors not look at lumbar vertebra: The CT work in this study was part of an atherosclerosis study of the thoracic cage- so lumbar vertebrae that are the most diagnostic of bone loss impairment we not included in this study.

This was explained in the original manuscript: lumbar vertebrae do not reliably appear in the CT image field of view. We only CT-scanned the thoracic cage, and not more of the body, to minimize radiation exposure for study participants.

It is relevant to know why Tsimane males in particular have low bmd and thoracic fracture – this is not an expected age-related phenomena. Is the low bmd a sign of another causative pathology that is causing the impairment of activity and pain? If so, low bmd is only a secondary measure of an underlying impairment that may not have to do with bone health at all – maybe dietary, parasitic, related to blood disorder that causes bone loss and impairment? Is this not relevant to discuss or mention?

With respect to BMD, we do not know what reviewer #2 is referring to when he/she states that “Tsimane males in particular have low bmd”. Low relative to what? Actually, Tsimane males do not have lower BMD compared to age-matched Los Angeles males (see Figure 2 in Stieglitz et al., 2019), suggesting that the former do not experience greater risk than the latter of more generalized skeletal fragility. Tsimane males do, however, show higher thoracic vertebral body fracture prevalence (see Figure 3B in the Stieglitz et al., 2019 *eLife* paper), which might result from high levels of mechanical stress on men’s vertebrae from frequent heavy load carrying. In the Discussion of the present manuscript we elaborate on potential causes of skeletal fragility for Tsimane, though we consider this issue at length in our other published papers (https://elifesciences.org/articles/48607; https://asbmr.onlinelibrary.wiley.com/doi/full/10.1002/jbmr.2730; https://onlinelibrary.wiley.com/doi/abs/10.1002/ajpa.23214; https://onlinelibrary.wiley.com/doi/full/10.1002/ajpa.22681). We elected not to focus more on fracture etiology in the present manuscript because of its focus on potential economic consequences of compromised bone strength.

What is going on with male bone loss and female bone loss is clearly different, as discussed by authors that point to previous studies that show female bone loss is related to post-menopausal or reproductive reasons. And both sexes show different "costs" of the bone health measure as API – they sexes needed to be discussed differently – and again why the “cause” of the male bone loss is important to explain. They likely have different etiology in both sexes, different costs associated because of this.

The revised Discussion addresses some of these sex differences. But as revised Figures 2-3 show, age-associated BMD declines in thoracic vertebral bodies might not be so disparate between sexes (in terms of intercepts and slopes), and we prefer to refrain from mischaracterizing etiology and excessive speculation.

Again wide sweeping statements with no basis for any meaningful interpretation: "Sixty-eight percent of adults age 50+ are also clinically diagnosed with some musculoskeletal problem by Tsimane Health and Life History Project (THLHP)" and "some physical discomfort during activities". – what is "musculoskeletal problem" and how can we use this as an equivalent of the BMD measure used in this study?

We have clarified these statements and we thank reviewer #2 for their attention to detail.

Statements that are not true or well researched/cite:"Ample bioarchaeological evidence of partially or fully healed fractures (e.g. (Lovejoy and Heiple, 1981; Pfeiffer, 2012)) and developmental abnormalities (Cowgill et al., 2015; Trinkaus, 2018) suggests longer-term survival with varying levels of disability.” In fact, numerous bioarcheological studies have shown that “osteopenic” fractures with low BMD are very rare in the past – traumatic/violent fractures are very common

We have since clarified and expanded upon these statements.

"Tsimane thoracic vertebral fractures appear to result from both trauma and senescence (Stieglitz et al., 2019) and are not just prevalent among the oldest adults.”So are we looking at violence or accidental trauma fracture in males here – or related to another primary disease? This needs to be discussed/expanded. It is not senescence in young age males.

We have since clarified and expanded upon these statements.

Again – both sexes -age groups have different etiologies of low BMD and likely different outcomes/costs because of this.Reviewer #3:The submitted manuscript describing productivity loss linked to bone mass decline is well written, novel and pertinent in several domains. It is certainly a worthwhile contribution. The suggestions made here for adjustments and improvements reflect this reviewer's responses to the work. The clear statements at beginning and end regarding the importance of this kind of research for bioarchaeological and paleoanthropological studies is welcome.Perhaps the broadest concern with the current version of the manuscript is the tendency to under-play the imbalance of impact of bone mass loss between men and women. Clearly, the paper's focus is on hunting and fishing, which is all good. However, from Abstract to conclusions the different pattern in women is underplayed. The type of loom(s) used by the weavers could be pertinent to the study and should be specified.

We have included a new figure (Figure 1) showing the actual looms used by women, and the finished woven products.

The brief section on the lack of significant effect on weaving should be moved up so that it precedes the section on long distance walking.

Done.

The relative inattention to women's productivity is a related concern.

We agree that this relative inattention to women’s direct production is a major study limitation.

There should be acknowledgment, as the authors discuss the limitations of the work, of the fact that their data are at least somewhat inconsistent with modern industrialized countries' clinical data, with regard to sex patterns in BMD and the disability load associated with compression fractures.

Here we are not exactly sure what Dr. Pfeiffer has in mind when she states that the Tsimane data are “at least somewhat inconsistent with modern industrialized countries' clinical data, with regard to sex patterns in BMD and the disability load associated with compression fractures.” Revised Figures 2-3 show linear age-related BMD declines for both sexes, as is typically observed in modern industrialized nations. Moreover, we now mention in the Discussion a previous study of Eekman et al., 2014, albeit using slightly different methods that are not directly comparable to those we use, that estimates costs of clinical spine fracture in terms of work absenteeism among osteoporotic adults aged 50+ years in the Netherlands. That study reports a cost of 25% of the mean annual per-capita income, which actually is within the range of productivity losses associated with fracture and lower BMD that we report, depending on whether one uses as a reference mean daily per-capita energy intake [cost of 15%] or intake from protein and fat [cost of 44%].

The research presented here links to a research area of which the authors appear to be unaware. In a 1996 paper (Anatomical Record 246:4:423-432), Agarwal and Grynpas asserted that osteoporotic fracture is absent in past populations. While it has only been cited 41 times, those citations include recent work.

Thank you, we now cite this paper.

I urge the authors to become better informed on just how novel – and helpful – their work is, in a bioarchaeological context. In my opinion, the best explanation for the "pattern" reported in that 1996 paper is that no one ever systematically studies thoracic compression fractures.

This interpretation seems reasonable, and we now mention it in the revised manuscript. We thank Dr. Pfeiffer for her insight.

Nevertheless, there is a literature arguing that osteoporosis is a modern disease (cf. Kralick and Zemel, 2020).

We thank Dr. Pfeiffer for mentioning this relatively new paper, which we now cite.

With regard to which vertebrae form the basis of the dataset: If T7 or T8 form the starting point for the assessment, the technician might be providing bone mass input from groups as different as T5-T7 to T8-T10. These are very different parts of the thoracic column, from the bone mass perspective. This is a concern that should be addressed, if possible. The morphological assessment of T6 to T12 helps to give the reader more confidence in the data.

We now address this.

The discussion of the contributions of older group members would benefit from stretching beyond physical activities to intellectual assets, along the lines of D.J. Levitin's argument that older adults contribute experience and judgement. The impressive breadth of the framework constructed by the authors could and should be even broader and more interdisciplinary.

We definitely agree, and have in fact published on other kinds of contributions of older Tsimane adults (some of these papers are cited). Thanks for bringing this up. We now expand our Discussion of the contributions of older group members, and cite Levitin’s work.

[Editors’ note: what follows is the authors’ response to the second round of review.]

The following is a summary of the essential revisions, with the detailed reviewer comments appended below for expanded direction and explicit instruction:The authors need to 1) reduce statements of implied causation in the discussion (and also in the general setup of the paper regarding the possible causal links between bone fragility and disability) and fully acknowledge all possibilities and 2) carefully establish potential mechanisms for any proposed causality, even when reduced explicitly to speculation. Currently the discussion is written as if the correlation is directional and infers causation, but with so many unstudied covariates (since lower BMD is a function of aging) this is inappropriate. The appropriate null hypothesis for interpretation should be carefully considered. I support some hypothesis generation and speculation in the discussion but this should be clearly indicated as such, and clearly separated from the conclusive interpretations. Even as such, the potential mechanistic links here would need to be much more carefully detailed and supported given the reliance on indirect inference only and the inconsistency with existing literature. Please see the reviewer comments for explicit suggestions and details.

Thank you for this detailed summary of the reviews—we found it very useful. In the revised Introduction and Discussion we now: i) omit any statements implying that our study definitively establishes causation; ii) emphasize that our results may be consistent with several possible explanations, not all of which are explicitly considered here; and iii) include a new conceptual model (see Figure 6 and accompanying text) that identifies diverse pathways linking bone fragility and diminished productivity. As suggested by Dr. Perry we are careful in the revision to recognize speculation in the discussion as such.

Reviewer #1:This is a very nice study of associations between bone fragility, disability, and subsistence costs among a small-scale forager-horticulturalist society, the Tsimane of Bolivia. The study builds on a great recent paper in eLife by the same authors showing that the Tsimane (especially women) have lower average vertebral BMD values than might be expected given their high physical activity levels, as well as surprisingly high rates of vertebral fractures. In this study, they investigate the impact of low vertebral BMD and fracture incidence on disability (or, "adult physical impairment" as they call it) and ultimately the economic costs of such disability. The paper nicely frames the study within evolutionary theory and provides novel insights on the selective pressures that may have shaped human longevity and our species' propensity for age-related fragility. The sample analyzed is impressively large, particularly for a remote-living population like the Tsimane. The CT methods used to collect the bone data are the gold standard. The paper is clear and well written, and the interpretations of the results are all reasonable. Also, the figures are nice. Overall, I think this paper deserves consideration for publication in eLife, but I have a couple of concerns.First, in my opinion, the possible causal links between bone fragility and disability are not considered or described in enough detail. The only mention of such a link is: "Vertebral deformities are reliably associated with chronic pain…(no citation)" Do the authors have data on back pain? If so, the link between bone fragility and disability would be a lot stronger if they could show that bone fragility is indeed associated with chronic pain among the Tsimane. If not, then it remains possible that bone fragility and disability are only indirectly related. For example, perhaps in many cases bone fragility is just one manifestation of a more general somatic decline, and disability is engendered primarily by other aspects of the general phenomenon and not bone fragility per se. If the authors do not have pain data, I suggest they at least make a stronger argument for the link between bone fragility and pain/disability by citing prior research demonstrating such links.

We agree, in retrospect, that we did not describe in sufficient detail potential mechanisms (e.g. back or other pain or discomfort) linking bone fragility and reduced economic productivity. We now make these connections more explicit, as suggested by all reviewers. We do this in the Introduction and Discussion (also see new Figure 6); the Discussion includes a new conceptual model linking bone fragility and reduced economic productivity during aging. We also note throughout the paper that there are several possible explanations for the findings that we report. We added relevant citations, including to the sentence mentioned by Dr. Wallace. Lastly, we added that prior anthropological studies of impairment and foregone productivity are similarly limited in that they overlook specific mechanisms. Unfortunately we lack systematic data on back pain specifically, but we are now considering ways to analyze existing medical data to obtain indirect insights into experiences of pain.

Second, on the same topic of causal links between bone fragility and disability, I believe the authors must provide a more detailed explanation for how they interpret the contribution of low BMD to disability. Independent of a fracture, there is no clear reason why one would expect low BMD to directly affect disability. Again, it is possible that low BMD and disability could be indirectly related if low BMD is only one of multiple outcomes of general somatic decline. But my understanding is that osteoporosis is generally an asymptomatic disease until fracture occurs. My impression is that the authors may not necessarily be arguing that low BMD, independent of fracture, affects disability, but rather that fracture plus low BMD leads to more disability than a fracture alone. Even so, why does this scenario make sense? Are fractures experienced by a bone with low BMD inherently more painful? Maybe the cooccurrence of fracture and low BMD leads to more disability because such fractures are generally more severe. Or, given that bone fractures can trigger bone loss, maybe the cooccurrence of fractures and low BMD represent cases where fractures took place longer ago and thus potentially have a greater effect on chronic pain and disability. An extreme way of resolving this uncertainty would be for the authors to disregard the BMD data entirely and focus just on the fracture data. The overall storyline would not change too much. However, I am not suggesting the authors do this. The BMD data are potentially important. But the links with disability need to be more carefully considered.

We appreciate Dr. Wallace’s request for clarification on mechanisms, especially pathways linking lower BMD and functional disability. His interpretation is broadly consistent with ours, and we agree that in the prior manuscript we did not explain this in enough detail. We now elaborate on links between lower BMD and disability in the text and in Figure 6, which we hope clarifies these mechanisms.

Reviewer #2:This paper aims to estimate the magnitude of productivity losses associated with compromised bone strength among the Tsimane. The authors specifically test "whether a specific morphological characteristic which may indicate physical impairment, i.e. compromised bone strength, is indeed associated with diminished participation in routine subsistence tasks". They "identify economic correlates of compromised bone strength (i.e. thoracic vertebral body fracture and lower thoracic vertebral BMD) by examining involvement in four common tasks in foraging economies: hunting, tree chopping, tool manufacture (i.e. weaving), and walking long distances".The data presented in this paper are important and have the potential to make a valuable contribution to the literature, and the authors are to be congratulated for conducing important, unique, and valuable fieldwork. The questions they have posed in this paper are interesting and worthy of consideration. That being said, I have some concerns with the analysis, what it assumes, what it excludes, and the conclusions.The paper is set to test the "probability of completely ceasing to perform a given task increases with compromised bone strength", and estimate the "magnitude of productivity losses (i.e. lost kcals/day) from hunting cessation associated with compromised bone strength" The authors claim "We do not assume that compromised bone strength is itself a proxy for functional impairment or reduced productivity. Instead we posit that individuals with compromised bone strength who are no longer able to perform specific subsistence tasks facultatively adjust their time allocation based on various factors (e.g. phenotypic condition, household demand) to maximize productivity." This is really unclear and woolly. What, specifically then, are they predicting? If they don't think compromised bone strength influences productivity, why are they testing it, and why is the text replete with indications that they are interpreting this as a direct influence?

We agree that our original phrasing was unclear, and we have since clarified this. Specifically, we write:

“We do not assume ex ante that compromised bone strength per se is the only cause of disability or reduced productivity, or that these factors necessarily co-vary in some coordinated fashion—these are empirical questions partially addressed in this paper. We do expect that individuals with compromised bone strength who are no longer able to perform specific subsistence tasks will adjust their time allocation depending on various factors (e.g. own somatic condition, household demand).”

We also hope that the new Figure 6 will clarify this point; we have also omitted opaque language throughout the manuscript on this point.

The authors make their assumptions of the direction of this relationship clear throughout the paper, through the use of language such as "To estimate men's hunting productivity losses associated with fracture and lower BMD". The paper is replete with the inference of this relationship, and while the authors might carefully choose their words in places to try to avoid causation, the inference is there for the reader to make. Perhaps another way of putting this is the following: the authors show correlations between measures of later life activity/productivity and both increase fracture prevalence and lower BMD. There are three parsimonious explanations for this correlation:1) The loss of bone quality is the cause of a decline in involvement in economic activity2) The loss of bone quality is correlated with age, which independently corresponds with later life declines in economic activity, and the two are not directly related but collinear3) A later life decline in activity is the cause of a decline in bone quality.

We thank reviewer 2 for their suggestion to describe in more detail different interpretations about statistical relationships between bone fragility and reduced economic productivity. We also thank reviewer 2 for outlining potential scenarios. We agree with reviewer 2 (and with reviewer 1) that unpacking the relationships by more explicitly considering mechanisms merits greater attention. As we mention above in our response to reviewer 1, we now describe in more detail potential mechanisms in the Introduction and Discussion, and alternative explanations that may be consistent with observed statistical relationships.

These are simplistic explanations, and there are likely other factors involved (see comments below), but for the purposes of simplicity, let's consider the above. The language of the paper is only oriented around the first explanation and the direction of causality is assumed through most of the text, but only really directly acknowledged when they briefly address the possibility of “reverse causality” (reductions in activity causing a reduction in bone strength). They attempt to test for “reverse causality” through indirect methods: comparing fracture status to self-reported reasons for task cessation, and the use of GLM to compare BMD of those who cease versus those who continue participation. I'm not convinced that these adequately address the challenges of the direction of the relationship. Self-reported causes may, in some cases, related to injury (particularly in severe cases), but are much more likely to relate to other factors entirely, and to be unrelated to BMD.

This is a valid point, and we repeatedly emphasize that these analyses are not direct or definitive tests of causality. We added a figure (Figure 6) to show multiple potential pathways linking somatic condition and economic productivity.

The GLM comparisons of BMD among groups defined by continued participation in activity may be influenced by study design, and I am not certain it is testing the reverse causation that the authors suggest.

We are not sure that we understand what is meant here by “comparisons of BMD…may be influenced by study design”. We have no evidence of specific sample biases, which we now clarify in the text (also see Appendix).

The most significant influence on later life bone mass is earlier life activity. Peak bone mass is commonly achieved between the ages of 20 and 30, and is the single most important predictor of later life bone mass. The more you build early on, the more bone you have to lose. Activity at age 20 may thus be stronger correlate of bone mass after the age of 40 than recent activity patterns. It is curious that the authors point out that "Physical strength peaks in the mid to late 20s, then declines continuously thereafter, reaching ~60% of maximum adult capacity by age 70 (Gurven et al., 2006)." This is essentially the same trajectory of bone strength acquisition and loss, and there is considerable evidence that bone strength corresponds with muscle induced loads. One could argue the most parsimonious explanation is that the most active and economically productive individuals during young adulthood have the highest peak bone mass, and are more likely to remain productive (or with higher than average productivity) later in life.

This is an excellent point, which we now acknowledge in the Discussion including in the new Figure 6.

Apart from the issues noted above, I remain sceptical about causation. There is, in theory, a direct causal relationship between fractures and decline or cessation of economic activity. This is supported by the relatively strong results, for example, the declines in weaving and ability to walk for a day associated with earlier fractures. What is the causal mechanism for the relationship with BMD? Going back to the three possible interpretations I mentioned earlier, I would have no issue with the statement of a hypothesis that vertebral fracture is the cause of a decline in involvement in economic activity.

Thank you for this suggestion, we now acknowledge this explicitly, as follows: “Human thoracic vertebrae track mechanics of both lower and upper limbs, and thoracic vertebral fracture can directly impede mobility.”

If I were to hypothesize what would explain the correspondence with BMD, I would go for a variant of the second hypothesis, that it is correlated with age, which independently correlates with other measures that are associated with later life declines. There are multiple collinear biological trends associated with senescence. While the authors provide no consideration of a biological mechanism through which BMD would be associated directly with productivity, the inference of causation is made without consideration of the particular relationship of declines in BMD associated with other factors of aging.

We agree, and the fact that both reviewers 1-2 converged on this suggestion convinces us that this was an important shortcoming of the prior submission. For this reason we included a new conceptual model (Figure 6) and text describing potential pathways linking bone fragility (including BMD) and diminished productivity. We emphasize repeatedly that our results may be consistent with several possible explanations, not all of which are explicitly considered here.

I am also sceptical of the consideration of both BMD and trauma as indicators of bone strength. BMD is highly correlated with bone mechanical function, this is uncontroversial, but considering trauma as a similar indicator may be problematic. Demographers and epidemiologists have often noted that trauma occurs in different societies with a relatively consistent frequency (at least when compared to infectious or metabolic disease, for example), because they result from accidental injury which tend to occur with some statistical regularity in larger populations. The data in Figure 2 suggest that there is no real difference the relationship between BMD and Age between groups of men with and without fracture, which could be interpreted as evidence that fracture prevalence among is predominantly determined by accidents rather than a loss of bone strength, regardless of the age or BMD of the individual. In contrast, there appear to be considerable differences among the women measured, suggesting that BMD and Fractures are more strongly correlated in women. The male data, at least, would seem to contradict the assertion that fracture prevalence is a proxy for bone strength.

It is fair to say that some fractures result from trauma, and some from fragility, and we are unable to determine the relative etiological contribution of each. We agree with reviewer 2’s general point that it would be problematic to conflate trauma with bone mechanical function. We also agree that Tsimane results suggest that men’s fracture may disproportionately result from trauma, including accident-induced trauma (as we acknowledge in the text; in Figure 6 and in a separate paper: https://elifesciences.org/articles/48607). These observations do not, in our view, contradict the idea that a fractured bone is more fragile (“strength-compromised”) than a non-fractured bone, all else equal. This idea is supported by studies showing that fractures provoke systemic bone losses, incomplete recovery of bone quality, and increased risk of future fractures and longer-term disability. Reviewer 2 is correct in stating that the association between BMD and fracture is stronger for Tsimane women than men. Men’s BMD is not significantly associated with fracture, defined as grade ≥1 on the Genant scale (i.e. ~20–25% reduction in vertebral height, and a 10–20% reduction in area). However, men’s BMD is inversely associated with borderline vertebral deformity, defined as grade 0.5 on the Genant scale (i.e. <20% reduction in height; for details see Appendix 1—table 17 here: https://elifesciences.org/articles/48607). For all of these reasons, we do not think that results for men invalidate the use of fracture status to ascertain degree of bone fragility.